# Transcending Cost-Quality Tradeoff in Agent Serving via Session-Awareness

**Yanyu Ren**[1]     **Li Chen**[2,*]     **Dan Li**[1,2]     **Xizheng Wang**[1]
**Zhiyuan Wu**[1]     **Yukai Miao**[2]     **Yu Bai**[2]

[1]Tsinghua University    [2]Zhongguancun Laboratory

{ryy23, wang-xz22, wu-zy25}@mails.tsinghua.edu.cn
tolidan@tsinghua.edu.cn   {lichen, miaoyk, baiyu}@zgclab.edu.cn

## Abstract

Large Language Model (LLM) agents are capable of task execution across various domains by autonomously interacting with environments and refining LLM responses based on feedback. However, existing model serving systems are not optimized for the unique demands of serving agents. Compared to classic model serving, *agent serving* has different characteristics: predictable request pattern, increasing quality requirement, and unique prompt formatting. We identify a key problem for agent serving: LLM serving systems lack session-awareness. They neither perform effective KV cache management nor precisely select the cheapest yet competent model in each round. This leads to a cost-quality tradeoff, and we identify an opportunity to surpass it in an agent serving system.

To this end, we introduce AGSERVE for **AG**ile **AG**ent **SERV**ing. AGSERVE features a session-aware server that boosts KV cache reuse via Estimated-Time-of-Arrival-based eviction and in-place positional embedding calibration, a quality-aware client that performs session-aware model cascading through real-time quality assessment, and a dynamic resource scheduler that maximizes GPU utilization. With AGSERVE, we allow agents to select and upgrade models during the session lifetime, and to achieve similar quality at much lower costs, effectively transcending the tradeoff. Extensive experiments on real testbeds demonstrate that AGSERVE (1) achieves comparable response quality to GPT-4o at a 16.5% cost. (2) delivers $1.8\times$ improvement in quality relative to the tradeoff curve.

## 1 Introduction

Large Language Model (LLM) agents are revolutionizing task execution by incorporating agentic workflows and refining responses through observational feedback from their surroundings [47, 27]. Gaining tremendous attention from both academia and industry, the application scenarios of LLM-based agents range from query agents that automatically extract information from large databases [39] to embodied agents that carry out household activities following user commands [45, 60].

Despite the rapid development of LLM agents, the serving systems powering them are largely dependent on existing LLM serving paradigms, which are not optimized for agent workflows. In a typical LLM serving process, the model takes in human instructions and performs complex reasoning to generate responses that satisfy human requirements. Compared to classic LLM serving, our analysis shows that *agent serving* exhibits distinct characteristics:

- **Predictable and High Frequency Interactions.** Agents utilize lightweight plug-ins to generate observations and initiate new requests at intervals of milliseconds [31, 29] after obtaining the response. Additionally, the responses to agent requests are usually short (within 100 tokens). The generation latency is also short and relatively constant.

---

*Corresponding Author

39th Conference on Neural Information Processing Systems (NeurIPS 2025).

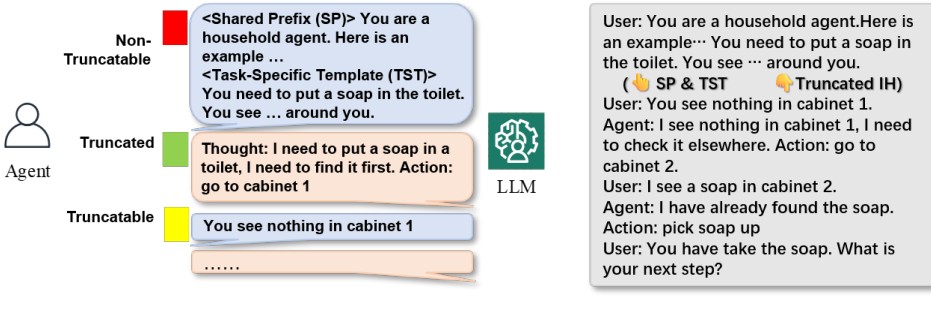

**(a) Session History Pattern**   **(b) Prompt after Truncation**

**Figure 1:** A prompt example in agent serving. LLM interacts with the agent in multiple rounds. The shared prefix (SP) and task-specific template (TST) are non-truncatable. The interaction history (IH) is truncatable. (b) shows the actual prompt after the first entry in IH is middle-truncated.

- **Growing Contexts and Increasing Difficulties.** Agents engage in multi-round sessions, and their contexts grow rapidly [56, 55] by the number of rounds. The task difficulty for LLMs also increases with longer context.
- **Unique Prompt Formatting.** Agents format prompts in unique patterns, such as middle truncation instead of prefix truncation, as shown in Fig 1.

Existing systems perform suboptimally in agent serving. Single-model approaches [61, 18, 11, 38] tend to overkill tasks in the early stages and underperform in the later ones. Routing-based methods [37, 13] and model cascading systems [7] suffer from inaccurate model selection and frequent model migrations, respectively. Consequently, they inherit the inherent *cost-quality tradeoff* from conventional LLM serving: achieving higher-quality responses requires interaction with larger models, which leads to increased costs—especially when using commercial APIs.

Looking into agent serving characteristics, we identify an opportunity to transcend the cost-quality tradeoff. We discover that the session predictability can be exploited to reduce the inference latency, and that the model of different sizes can be applied in different segments of the session lifetime to reduce costs. Thus, we conclude that to transcend the cost-quality tradeoff, we can leverage *session-aware* KV cache reuse and eviction policies together with *session-aware* model cascading.

Therefore, we are motivated to design an agent serving system to improve agent serving efficiency and quality. However, realizing this vision requires addressing three key challenges:

**C1 Session-Aware Cache Policy.** Modern serving systems employ reuse and eviction policies to reduce the recomputation overhead across requests. However, existing serving systems enforce strict prefix-matching for reuse [19] and adopt simple Least-Recently-Used (LRU) eviction policies [24, 62]. Neither policy fits well in agent serving, which leads to low KV cache hit rates and undermines efficiency in heavy agent-serving scenarios.

**C2 Quality Assessment.** Existing systems overlook session lifetime and lack mechanisms to dynamically assess model response quality under long-context conditions. As a result, they often overcommit large models too early, incurring extra costs, or underperform in later stages due to poor model choices. An effective agent-serving system must not only select a good initial model to reduce early-stage migration, but also assess response quality in real time to inform upgrades.

**C3 Resource Allocation.** The model switching imposes dynamic and imbalanced resource demands across models. Static GPU allocation wastes resources, while existing systems do not adaptively provision GPUs based on workload shifts. Thus, the serving system must support fine-grained, dynamic resource allocation to ensure high overall GPU utilization.

To address these challenges, we design AGSERVE, a session-aware cascading agent serving system. Specifically, to address **C1**, AGSERVE introduces in-place positional embedding calibration to enable high KV cache reuse under middle-truncation. It further improves cache efficiency via the novel cache eviction policy based on the estimated time of arrival (ETA). To address **C2**, AGSERVE integrates a Quality Maintenance Module (QMM) to assess task difficulty and response quality in real time, allowing informed model upgrades throughout session lifetime. To address **C3**, AGSERVE implements a resource scheduler that dynamically allocates GPU resources based on supply-demand statistics. AGSERVE also exposes a set of APIs to facilitate flexible integration by agent developers. By jointly optimizing KV cache reuse, model selection, and resource allocation, AGSERVE transcends

the traditional cost-quality tradeoff in agent serving. It achieves lower latency and reduced serving cost while maintaining comparable response quality.

We implement and deploy AGSERVE on real testbeds, and evaluate its performance using Agent-Bench [29]. Results show that, with similar quality to GPT-4o, AGSERVE reduces agent serving cost by 83.5%. At the same cost, AGSERVE can achieve $1.8\times$ higher quality relative to the cost-quality tradeoff curve. For multi-agent workloads, AGSERVE reduces cost by 64% and achieves $1.6\times$ quality compared to existing LLM serving systems. AGSERVE also raises KV cache hit rate by $2.84\times$ and reduces the round duration by up to 50% with session-aware KV cache management. It also accelerates agent serving by $1.2\times$ with the dynamic allocation strategy.

Our work is available at `https://github.com/robinren03/agserve`.

## 2 Understanding Agent Serving

### 2.1 LLM Agent Preliminary

LLM agents often interact with the model using a structured prompt consisting of a *shared prefix*, a *task-specific template*, and references to available *tools*, allowing them to iteratively carry out the *task*. The shared prefix explains tools and gives few-shot examples [5], while the task template gives the specific task. In each round, the agent requests LLM's chain of thought (CoT) [54] response, performs the action in it, and appends observations. Due to context length limits, LLM agents may truncate the interaction history but preserve the SP and TST to maintain correct behavior.

### 2.2 LLM Serving Systems are Insufficient for Agent Serving

To better understand the motivation behind the design of AGSERVE, we first examine the key differences between classic LLM serving and agent serving.

We exhibit their **workload differences** in Fig 2. Mainstream LLM serving systems provide inference service of a certain model [24, 21]. They schedule and run the sequences in each node with no quality concerns. To our knowledge, *there is currently no system specifically designed for agent serving*. We argue that agent serving systems should go beyond conventional LLM serving systems since their task is to provide quality-guaranteed actions. They incorporate an additional transparent layer to choose an LLM from a pool, and then also perform request routing and scheduling.

Their **behavioral differences** lead to poor performance of existing systems.

**B1** Agent serving exhibits frequent and regular request patterns in each round. As shown in Fig 3(a), the token number grows like a staircase across rounds. The arrival time for the next request under the same session is predictable [50]. Compared with human-oriented LLM serving [44], Fig 3(b) reveals that agent requests have shorter output lengths and shorter request intervals. These imply that agent serving constitutes a *heavy-prefill, light-decode* workload. Existing prefill-decode disaggregation systems such as MoonCake [41] and DistServe [63] are optimized for decode-intensive tasks, which brings little benefit for agent serving [21].

**B2** Agent sessions continuously grow by appending new responses to IH, while enforcing middle truncation if needed. Existing systems employ prefix-based cache reuse [19], which assumes strictly matched prefixes. Middle truncation breaks the prefix pattern and renders techniques like CachedAttention [17] ineffective. In parallel, existing systems such as vLLM [24] and SGLang [62] apply LRU policy to manage cache eviction. LRU is unaware of session semantics and may evict session caches that are about to request, further incurring a low cache hit rate.

**B3** As agent sessions progress, the task complexity increases with a growing demand for larger models. Fig 4 captures this trend with a dropping satisfactory rate on one model. Smaller models tend to generate invalid actions or violate preset rules as shown in appendix §B. Existing routing [37] and cascading [13] systems overlook this trend by treating each request independently, leading to fragile model choices and costly migrations.

## 3 AGSERVE Overview

**AGSERVE Architecture.** As presented in Fig 6, AGSERVE comprises three pieces: server, client, and scheduler. We describe the roles and submodules of each of these components as follows,

- The **Session-Aware Server (SAS, §4)** manages session caches and performs LLM inference. SAS maintains a Session ID-Sequence Table (SIST), which maps session IDs to their latest sequences. When a new request from an existing session arrives, SAS extracts its session cache from memory and performs in-place positional embedding calibration if needed. SAS maintains the ETA-based cache eviction (ECE) policy in case of GPU memory overloads.

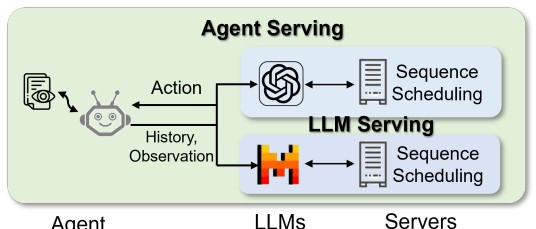

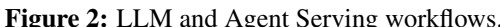

Figure 2: LLM and Agent Serving workflows.

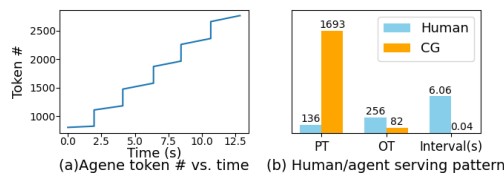

Figure 3: Comparison of LLM and agent serving behaviors. *PT/OT* represent the prompt/output token numbers, tested on our A6000 testbed.

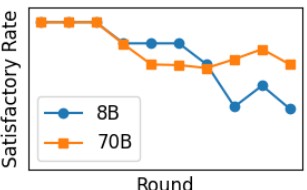

Figure 4: Agent serving satisfactory rate drops as round increases.

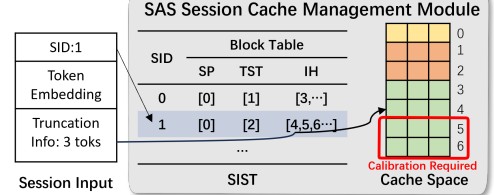

Figure 5: SAS session cache management module.

- The **Session Guard Client (SGC, §5)** provides observations for the agent and ensures the serving quality throughout the session. The client monitors the lifetime of the session and determines the LLM adoption. To achieve this, SGC employs a Q-Judge to select the most cost-efficient model for the session. It also runs a daemon R-Judge to periodically check response qualities, deciding mitigation actions if quality issues arise.
- The **Resource Scheduler (RS, §6)** assigns LLMs onto GPUs and dynamically adjusts the configurations based on the real-time demand-supply ratio across the system.

**AGSERVE Workflow.** When a new agent starts, the SGC assesses the task's difficulty with Q-Judge. SGC establishes a session with an SAS instance or external API that provides service for the given model. Within the session, the agent then interacts with the LLM for multiple rounds.

Throughout the session, SGC monitors the reasoning quality using the R-Judge. If performance falls below a pre-designated threshold, the SGC initiates a retry or requests a migration to a more powerful model instance. In the case of a migration, SGC terminates the current session and builds a new session with a SAS or API of a more powerful model. Furthermore, the RS looks at the requests that each model handles and dynamically adjusts the GPU resource distribution across models.

## 4 Sessionizing KV Cache to Maximize Cache Hit in SAS

AGSERVE adopts PagedAttention following the practices of vLLM and SGLang. We incorporate SIST into the block table as shown in Fig 5. SIST introduces session awareness to the KV cache by maintaining each session's block ID. Our two following innovations take advantage of SIST.

**In-place KV Cache Calibration.** Modern LLMs adopt Rotary Positional Embedding (RoPE, [46]) to embed tokens. CachedAttention [17] proposes decoupled KV cache. However, the decoupled cache consumes twice the space, one for the decoupled KV cache to preserve and one for the normal cache to decode. Instead, AGSERVE adopts in-place calibration to facilitate middle truncation. The in-place calibration multiplies the original key cache by $e^{i \cdot \delta_p \theta}$ where $\delta_p = p' - p < 0$ is the opposite

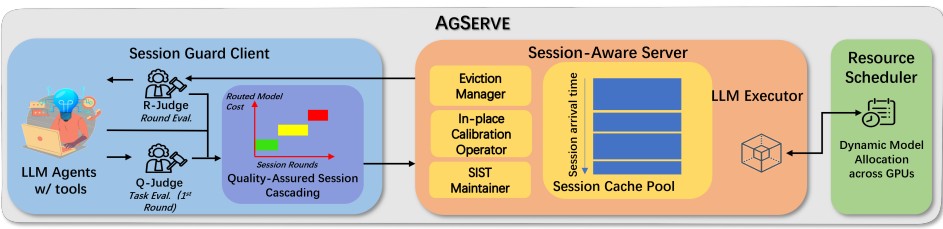

Figure 6: AGSERVE Architecture and Workflow.

**Figure 7 illustration**

| | Time | 0:03 | 0:04 | 0:05 | 0:06 | 0:07 (Unexpected Behavior) | 0:08 | 0:09 |
|---|---|---|---|---|---|---|---|---|
| | Request SID | 3 | 0 | 1 | 2 | 0 | 3 | 1 |

LRU (vLLM, SGLang,⋯) — Cache Hit: 1/7

| Time | SID / LRU time | | |
|---|---|---|---|
| 0:03 | 2 / 00:03 | 1 / 00:02 | 0 / 00:01 — Miss |
| 0:04 | 3 / 00:04 | 2 / 00:03 | 1 / 00:02 — Miss |
| 0:05 | 0 / 00:05 | 3 / 00:04 | 2 / 00:03 — Miss |
| 0:06 | 1 / 00:06 | 0 / 00:05 | 3 / 00:04 — Miss |
| 0:07 | 2 / 00:07 | 1 / 00:06 | 0 / 00:05 — Hit! |
| 0:08 | 0 / 00:08 | 2 / 00:07 | 1 / 00:05 — Miss |
| 0:09 | 3 / 00:09 | 0 / 00:08 | 2 / 00:07 — Miss |

**4x Improvements**

ECE (ours) — Cache Hit: 4/7

| Time | SID / ETA time | | |
|---|---|---|---|
| 0:03 | 0 / 00:05 | 1 / 00:06 | 2 / 00:07 — Miss |
| 0:04 | 0 / 00:05 | 1 / 00:06 | 3 / 00:08 — Hit! |
| 0:05 | 1 / 00:06 | 3 / 00:08 | 0 / 00:09 — Hit! |
| 0:06 | 3 / 00:08 | 0 / 00:09 | 1 / 00:10 — Miss |
| 0:07 | 3 / 00:08 | 0 / 00:09 | 2 / 00:11 — Hit! |
| 0:08 | 3 / 00:07 | 2 / 00:10 | 0 / 00:13 — Hit! |
| 0:09 | 2 / 00:10 | 3 / 00:12 | 0 / 00:13 — Miss |

**Figure 7:** An illustration of how ECE outperforms LRU. LRU policy decides the evicted session cache based on the LRU time, while ECE is based on the ETA of the next request from the session. In this simplified demonstration, there are four sessions ongoing with no shared blocks. The SAS can only preserve three in the session cache. All sessions share the same arrival pattern, and thus, their request arrives at SAS in a round-robin fashion. LRU policy, adopted by vLLM and SGLang, evicts the cache of sessions to arrive, causing a low cache hit rate. On the other hand, ECE achieves a session cache hit rate of 75% in theory. We also consider an occasional unexpected behavior at 0:07, causing the reversed order of 0 and 3. ECE still performs robustly with a consistent hit rate. Notably, the ECE policy does not change the ETA for occasional flapping. Taking all into consideration, ECE achieves a 4x hit rate compared to LRU under this case. In a more complicated case, ETA of different sessions from the processing time may differ, and sessions may share blocks.

number of truncated tokens. This operation calibrates the position embedding from $e^{ip\theta}$ to $e^{ip'\theta}$. In practice, SAS performs

$$
\begin{aligned}
K'_{p',2i} &= cos(-\delta_p\theta_i)K_{p,2i} + sin(-\delta_p\theta_i)K_{p,2i+1} \\
K'_{p',2i+1} &= cos(-\delta_p\theta_i)K_{p,2i+1} - sin(-\delta_p\theta_i)K_{p,2i}
\end{aligned}
\tag{1}
$$

to reuse the cosine-sine cache of the RoPE module, further reducing the computation time.

**ECE Eviction Policy.** SAS needs to evict KV caches due to limited memory in online serving. To formulate the question, we assume that there are $n$ sessions ranked by the arrival time of their pending requests $sc_0, sc_1, ..., sc_n$. SAS predicts the arrival time if the next request has not arrived yet. The cache size of session $sc_i$ is $l_i$. We take two steps to review SAS's ECE eviction policy.

First, we assume that *all session caches are of the same size*. Inspired by the OPT strategy [40] in CPU cache, we remove $sc_i$ in the order of $n$ to 1. We prove its optimality in appendix §A.2.

However, session cache sizes are different, meaning that evicting one session may not spare enough space for another session, and the above solution cannot achieve the global optimal. To tackle this, we introduce a new symbol $T(k)$ as the optimal total TTFT for only considering $sc_{n-k}$ to $sc_n$.

To spare enough space for the first $k$ pending sessions, we need to decide on a set $E$ of session caches to evict. The eviction of $sc_i$ costs $P \cdot l(sc_i)$ to recompute and affects the $n - i + 1$ sessions behind, leading to a penalty of $(n - i + 1)Pl(sc_i)$. Thus, $T_k(n) = \Sigma_{i \in E_k}(n - i + 1)P \cdot l_i + (n - k)D_k + T(n - k)$, where $D_k$ is the average decoding time consumption with $k$ requests handling in parallel. $P$ is the prefilling overhead of one token relative to the average decoding time cost, considered as a penalty for evicting the session cache. For each $k$, SAS finds the $E_k$ to minimize $T_k(n)$ with dynamic programming, and overlooks $T(n - k)$ to avoid excessive eviction. Finally, SAS finds the minimum $T_{\hat{k}}(n)$ among all $k$s and evicts $E_{\hat{k}}$. Each SAS predicts the ETA upon the request trace of sessions and makes the eviction decision based on the ETA. We discuss the detailed algorithm and efficiency for ECE policy in appendix §D.

SAS does not swap caches to CPU, since such swapping introduces high overhead with long contexts and blocks other sequences in the batch. SAS schedules each sequence in a First-Come-First-Served manner instead of SGLang's radix approach to avoid starvation.

## 5 Exploiting Session-Level Cascading to Minimize Cost in SGC

SGC leverages the Q-Judge to assess task difficulty and establish sessions for agents with capable models. The Q-Judge $QJ(t, p)$ takes the agent task $t$, and the observation $p$ as input. $QJ(\cdot)$ assigns a difficulty label among 0, 1, and 2 for each task. SGC leverages the R-Judge for quality monitoring.

## 5.1 Reducing Overkill and Underkill of the Task

Q-Judge inevitably faces deviation from the ground truth, causing *underkill* or *overkill*. In the case of underkill, SGC needs to migrate several times to reach the capable model, wasting SAS resources. If overkilled, SGC faces an irreversible cost. We observe that SGC prefers an underkill rather than an overkill. We optimize the loss function for $QJ(\cdot)$ as follows, with $p, q$ as the ground truth and predicted probabilities, respectively, and $g$ as the ground truth label.

$$f(p, q, g) = -\sum_{i=1}^{n} p(x_i) log(q(x_i)) + \beta \cdot \sum_{i=1}^{n} (x_i - g)^2 \cdot \frac{\alpha_i}{1 - q(x_i)} \tag{2}$$

in which $\alpha_i = 1$ if $x_i > g$ (overkill) and $0 < \alpha_i = \alpha < 1$ if $x_i < g$ (underkill), $x_i$ is the value of the $i$-th label. The first element of the function is the traditional cross-entropy loss. The second element gives extra loss to the wrong labels if it is far from or larger than the ground truth.

We train the judge on a customized Chatbot-Arena dataset [8]. Details are disclosed in appendix §C.

## 5.2 Real-Time Response Quality Monitoring and Issue Mitigation for Long Context

We observe four typical types of quality issues in agent serving, namely service failure, violation of preset rules, invalid actions, and low reasoning quality. The first three are easy to identify by algorithmic approaches. For the fourth issue, SGC calls the R-Judge classifier to evaluate the reasoning quality at a user-customized frequency of $\nu$.

**Designing and Training R-Judge.** The prompt length easily exceeds the context window of modern classifiers with over a thousand tokens. Thus, we only input selected elements from the context to reduce the workload. R-Judge $RJ_\theta(m, r, t, e)$ takes in four parameters. $m$ indicates the model size. It provides the R-Judge with a bias of the serving quality against the model size. $r$ is the thought part of the CoT response. If the implementation does not adopt CoT, R-Judge takes the whole response as $r$. $t$ is the task of the agent. $e$ is the latest actions and observations when AGSERVE sends the request. $RJ(\cdot)$ judges based on how well $r$ can help assist the task of $t$ in the condition of $e$, covering cases of evasive and redundant replies. $\theta$ reflects the strictness of the judge. $\theta = 0$ is the strictest, which always considers the response as disqualified, and $\theta = 1$ allows all responses to pass the test. Users can adjust this threshold based on their own quality requirements or workload characteristics.

We disclose the training details of R-Judge in appendix §C.

**Addressing Quality Issues.** SGC adopts *retry* or *migrate* to mitigate quality issues. The retry strategy reruns the response generation, taking advantage of the randomness. However, if the response still faces quality issues, SGC forces a service-upgrade migration. Migration routes the session to another serving instance. SGC implements two types of migration. The *service-upgrade migration* routes the session to a larger LLM instance or API. The *general migration* decides the model option, with the possibility of a smaller model in case of service failures, such as timeouts or network issues.

SGC also supports restoration for certain agents. Restoration reverts the session to the last checkpoint, shortening the chat history to reduce reasoning complexity and improve performance.

# 6 Model Allocation across GPUs to Maximize Hardware Utility in RS

Efficient model allocation is crucial for optimizing resource usage in agent serving systems. AGSERVE employs a dynamic allocation strategy to ensure high resource utilization.

**Model Instances.** We divide the models into two categories *adjustables* and *presets*. Adjustables fit in a single node, while presets must span across multiple nodes. RS confines adjustable instances to a single node to minimize communication latency. For each model $m$, RS profiles the minimum number of GPUs required for inference, denoted as $w_m$. Each instance utilizes a multiple of $w_m$ GPUs. RS relies on manual operation for presets, since they have heavy initialization overhead.

**Dynamic Allocation.** RS dynamically allocates models based on model demand and resource availability. RS tracks the frequency of inference requests across all instances for each model and calculates its demand by summing up the calling frequency. The scheduler determines the supply by the available tokens to preserve in the KV cache of all instances.

RS prioritizes models with a high demand-to-supply ratio, scaling up their instances while reducing instances for models with lower ratios. RS redistributes model instances across available nodes to optimize resource usage. RS consolidates instances of the same model wherever possible to leverage parallelism. For example, RS starts three instances, utilizing 1, 2, and 4 GPUs each if 7 GPUs of the same device are allocated for a model with $w_m = 1$.

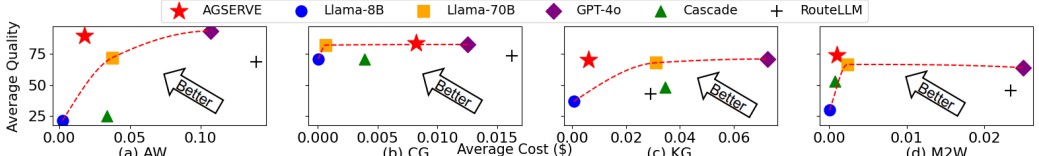

**Figure 8:** The red line shows the cost-quality tradeoff in each agent. AGSERVE breaks the cost-quality tradeoff in all four agents, achieving comparable quality to GPT-4o at lower costs.

## 7 Evaluation

We implement AGSERVE in Python with SAS based on vLLM [24]. We also implement customized CUDA kernels to support batched in-place KV cache calibration. We adapt four agents (AW,CG,KG,M2W) with AGSERVE's API. We disclose their details in appendix §E.1. We evaluate AGSERVE with these four agents and models of the 8B, 30B versions of Llama-3 and GPT-4o, due to generality of Llama structure and our testbed capability(see appendix §D). Without explicit benefit from the model selection, our key findings are

- With similar quality to GPT-4o, AGSERVE reduces serving cost by 83.5%.
- At the same cost, AGSERVE achieves 1.8× quality relative to the cost-quality tradeoff curve.
- Under multi-agent workloads, AGSERVE reduces cost by 64% and achieves 1.6× quality compared to an adapted state-of-the-art (SoTA) LLM serving system.
- AGSERVE achieves 2.86× cache hit rate with ECE policy than LRU, and reduces the round latency by up to 50% with in-place positional embedding calibration combined. AGSERVE accelerates agent serving by 1.2× with dynamic hardware resource allocation.

### 7.1 Testbed and Metrics

We evaluate the performance of AGSERVE based on AgentBench [29] over two testbeds. The first consists of two nodes, each equipped with four A6000 GPUs (48GB per GPU). The second comprises two nodes, each equipped with eight A800 GPUs (80GB per GPU), interconnected via PCIe.

Our evaluation focuses on three metrics of the sessions: end-to-end (e2e) latency, quality, and cost. Sessions that terminate due to invalid actions or other quality issues before task completion or reaching the round limit are excluded from latency measurements.

We rank each agent serving system with a quality score out of 100 concerning three aspects to follow the ranking mechanism of AgentBench,

- 25% for behaving normally until the task is completed or the interaction rounds limit is reached.
- 50% for the wellness of handling the task.
- 25% for completing the task in the fewest rounds possible, to benefit competitive model selection.

We disclose the details of the quality metric of each agent in appendix §E.2.

The cost includes two parts: the cost of nodes decided by the e2e latency of each agent task, and the cost of OpenAI APIs. For open-source models, we compute the cost based on the testbed capacity, selected strategy, and e2e latency. We provide details of model costs in appendix §E.3.

To exhibit AGSERVE's performance, we enlist a vLLM-adapted cache-centric strategy, and a model routing strategy, namely RouteLLM [37], as baselines for each experiment.

### 7.2 End-to-end Cascading Serving

For this evaluation, we disable dynamic allocation and run two instances on the A6000 testbed, one running Llama-8B and the other running Llama-70B.

**Baselines.** We compare the cascading strategy of AGSERVE to the following strategies.

- **vLLM+.** We adopt a single model throughout the serving process, like LLM serving systems. We pick one model among Llama-8B, Llama-70B, or GPT-4o each time, the same model choices in AGSERVE. We adopt prefix caching and a retry strategy to facilitate the approach.
- **Cascade.** Cascade adopts three-layer cascading, but disables the R-Judge and Q-Judge. It always starts at the smallest model and only addresses explicit quality issues.
- **RouteLLM.** RouteLLM trains four routers on the same Chatbot-Arena dataset as AGSERVE. Since matrix factorization and similarity-weighted ranking route almost all requests to the larger model, we adopt the BERT router with a preference for the smaller model at 0.8.

**Evaluation Results.** We exhibit the average cost-average quality curves of four agents in Fig 8. We utilize Piecewise Cubic Hermite Interpolating Polynomial [3] to draw the tradeoff curve. As

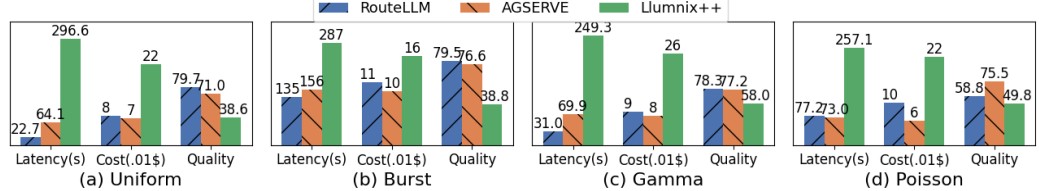

**Figure 9:** AGSERVE's performance in different distributions.

**Table 1:** Performance of routing models. S/M/L represents small/medium/large labels.

|  | S | M | L | Acc. | Low | High |
|---|---|---|---|---|---|---|
| GT | 112 | 306 | 173 | - | - | - |
| QJ | 190 | 397 | 4 | **70.4%** | **109** | 6 |
| RL | 374 | N/A | 185 | 57.9% | 161 | 88 |
| CE | 591 | 0 | 0 | 62.4% | 182 | **0** |

**Table 2:** Latency comparison between dynamic and static allocation strategies. Average, P90, and P50 exhibit the average, 90th percentile, and median of the e2e latency, respectively.

|  | Average(s) | P90(s) | P50(s) |
|---|---|---|---|
| AGSERVE | **119.38** | **177.05** | **106.81** |
| FS1 | 125.43 | 206.86 | 106.85 |
| FS2 | 242.35 | 295.86 | 237.93 |

displayed, AGSERVE breaks the cost-quality tradeoff in all four agents, achieving similar quality to GPT-4o with a 16.5% cost in AW. For the same cost, AGSERVE gets a 1.8× quality than the tradeoff curve. In comparison, without R-Judge, the Cascade strategy only significantly transcends the tradeoff in one of the four agents. On the other hand, RouteLLM performs below the cost-quality tradeoff in all four agents. We believe that this is largely due to two reasons. First, RouteLLM is designed for short inputs such as MMLU [20] and GSM8K [9], not for long-input agent serving. Second, RouteLLM is sensitive to the output of the routing model due to a lack of session-awareness.

### 7.3 End-to-end Multi-Agent Serving

In this section, we evaluate the multi-agent performance of AGSERVE in a distributed service scenario. We run the evaluation on our A800 testbed with a TP size constrained to 4 or below.

**Baselines.** We compare AGSERVE with two baselines.

- **Llumnix++ [48].** Llumnix is the SoTA LLM serving system to manage request allocation backed by vLLM. We extend Llumnix with support for Llama-3 and integrate Llumnix with AGSERVE's QMM. Llumnix++ runs four 4-GPU instances, two each for Llama-8B and Llama-70B, since Llumnix requires each node to run the same model.
- **RouteLLM.** We adopt the same route setting in §7.2.

**Datasets.** To our best knowledge, there is currently no real dataset on the traces of agent launching. We generate the traces of uniform and burst distributions. Similar to LLM serving [61, 28, 48], we also use Poisson and Gamma distributions to synthesize request traces. The trace only defines the launch time of each agent and does not affect any following interaction rounds in the session.

**Evaluation Results.** As shown in Fig 9, Llumnix++ has higher latency and lower quality in the high-frequency agent serving scenario. AGSERVE reduces cost by 64% and, in the meantime, achieves 1.6× quality compared to Llumnix++. RouteLLM routes most of the requests to GPT-4o and utilizes large and cheap computing resources to achieve low latency, and has a high latency in bursts due to the rate limit. With a much more expensive hardware rent, AGSERVE achieves comparable quality with only 81% of RouteLLM costs, and can reach 40% cost with wholesale-price cloud service.

### 7.4 Ablation Studies and Microbenmarks

### 7.4.1 Performance of SAS Session Cache

We evaluate the performance of session caching on agent serving. We deploy a Llama-8B instance on one A6000 node. The temperature is set to 1, following the OpenAI API default.

**Correctness.** We compare AGSERVE against vLLM, and exhibit the latency and quality of four agents in Fig 10. Latency is shown per request for CG due to its nondeterministic response and varying rounds. AGSERVE achieves up to 2× acceleration as in AW. The quality scores between AGSERVE and vLLM show a negligible discrepancy with no correctness concerns.

**Efficiency.** We evaluate the hit rate and TTFT for AW agents on Llama-8B SAS with limited memory and context window size. Fig 11 exhibits a 2.86× hit rate in the batch size of 8 and 2.80×, 2.14×,

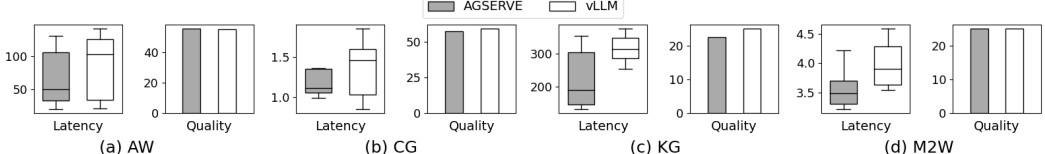

**Figure 10:** Latency (in seconds) and quality of vLLM and AGSERVE SAS across different agents.

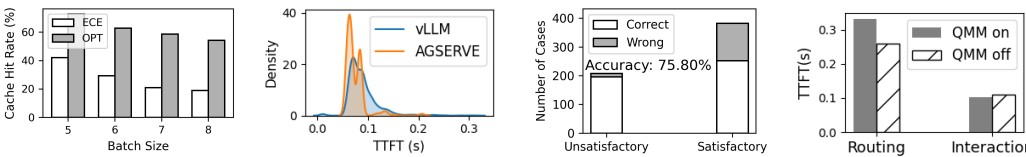

**Figure 11:** Cache hit rate by block.

**Figure 12:** KDE plot of TTFT.

**Figure 13:** R-Judge output against ground truth.

**Figure 14:** TTFT with and without QMM.

1.73× in the batch sizes of 7, 6, 5, respectively. Fig 12 shows the Kernel Density Estimation (KDE) plot of the TTFT. Benefiting from the high cache reuse by ECE eviction (peak around 0.06s) and in-place calibration (peak around 0.08s), TTFT of AGSERVE is significantly to the right of vLLM, showing less prefilling latency.

### 7.4.2 Performance of SGC QMM

We evaluate QMM to ensure proper quality maintenance with low overhead.

**Correctness of R-Judge.** To our knowledge, there is no existing human preference data for agent serving. We ask experts to label the responses of different-sized models, spanning across all four agents. We depict the performance of R-Judge in Fig 13, with a **95.2%** recall of unsatisfiable results. We exhibit the false positive and false negative rate of the R-Judge when picking different $\theta$ in Table 3.

**Correctness of Q-Judge.** It is hard to label the difficulty of prompts. We utilize the dataset above and build a loose restriction on the label given. For example, if the medium model response is satisfiable, it is considered right to label it as both small and medium. We compare Q-Judge(QJ) with RouteLLM(RL) and a model trained by the cross-entropy loss(CE) in Table 1. Q-Judge achieves high accuracy and few predictions that are higher than the ground truth, as we require in §5.1.

**Efficiency.** In real-world applications, AGSERVE makes the most use of the QMM's context windows by filling them up with session history. Fig 14 shows the influence of QMM on TTFT in the real application, when the average context length reaches 500 of the 512 window size. Still, QMM causes negligible overhead in normal interactions and an acceptable overhead when routing. We list their respective overhead in appendix §E.7.2.

### 7.4.3 Performance of RS Dynamic Allocation

RS only conducts reallocation for a significant imbalance in the demand-supply ratio across models to avoid excessive weight loading. We manipulate $\theta$ of the R-Judge to create such a scene. We evaluate the performance of AGSERVE on one A800 node, against two fixed allocation configurations. FS1 allocates four GPUs to Llama-8B and Llama-70B each; FS2 runs an 8-GPU Llama-70B instance. We disregard sessions that violate preset rules, accounting for roughly 6% of all sessions across FS1, FS2, and AGSERVE. We reveal the latency of all other sessions in Table 2. AGSERVE reduces 14% P90 latency compared to FS1. Compared to FS2, an LLM serving approach, AGSERVE achieves 49% average e2e latency. We provide a case study in appendix §E.7.3.

**Table 3:** False positive and false negative rates of the R-Judge under different $\theta$.

|  | 1 | 2 | 3 | 4 | 5 | 6 |
|---|---|---|---|---|---|---|
| False Positive Rate | 0.00% | 28.09% | 40.06% | 41.22% | 47.93% | 57.55% |
| False Negative Rate | 33.64% | 17.60% | 8.98% | 3.94% | 0.91% | 0.00% |

# 8    Limitations and Discussions

**R-Judge Sensitivity.** AGSERVE relies on R-Judge to perform cascading. When $\theta \equiv 0$, AGSERVE falls to GPT-4o, and $\theta \equiv 1$ degrades AGSERVE to Cascade as in §7.2. We show R-Judge's comparable performance to humans in appendix §E.8 despite the possible cost of early migration. We acknowledge that a judge trained with more sufficient data may benefit agent serving better.

**Node Routing.** SoTA LLM serving frameworks, such as SGLang [62] and MoonCake [41], decide the machine routing of each request by prefix matching. Due to the limited testbed size, AGSERVE binds each session to a specific instance (i.e., the most idle one at first-round scheduling). We recognize that there are two future directions to uncover: (1) the migration of in-flight sessions to balance load; (2) to exploit the prompt length predictability as in Fig 4(a) to match the instance.

**Large Scale Agent Serving.** Previous works [48, 42] suggest that a distributed scheduling outperforms in large-scale environments. We present a theoretical simulation in appendix §F about a distributed AGSERVE for large-scale serving and show AGSERVE's potential application.

**Broader Impacts.** The technologies used in AGSERVE make agent serving cheaper and more efficient, and do not bring extra societal impact beyond that of existing agents and LLMs. We rely on external safeguard measures such as red-teaming to minimize negative impacts.

# 9    Related Work

**Cache-centric LLM Serving.** Early-stage works such as SpotServe [34] and AlpaServe [28] mainly focus on parallelism strategies. Orca [61] proposes continuous batching and makes decoding memory-bounded. vLLM [24] improves KV cache management with PagedAttention. Further, Llumnix [48] explores multi-instance LLM serving with virtual memory and request migration. TetriInfer [21] proposes the idea of scheduling sequences according to length prediction, which is similar to the AGSERVE's session routing strategy. While TetriInfer makes scheduling decisions by the memory usage at present or shortly afterwards, AGSERVE looks into a resource usage function over time. Splitwise [38] and DistServe [63] disaggregate the prefill and decode in LLM inference. However, agent serving, a light-decode scenario, does not work well with disaggregation and its tools such as CacheGen [30], as we previously mentioned in §2.2. Quantization [16, 6] and kernel optimization [10, 52] can also accelerate inference. CacheBlend [58] selectively recomputes the KV caches for Retrieval-Augmented Generation, which is orthogonal to AGSERVE.

**Multiple-model serving.** Tabi [53] proposes automatic cascading using confidence, followed by FrugalGPT [7] focusing on the financial sector. AGSERVE uses prompt engineering to overcome the generality problem of FrugalGPT. HybridLLM [13] and RouteLLM [37] route requests to LLMs. Like other traditional LLM serving systems, they consider requests from the same session independently, making the results extra sensitive to the router without quality assurance. OnlineCascade [36] proposes an approach to learn cascading strategies, which is orthogonal to AGSERVE. Speculative decoding [26] uses small models to mimic the large ones with target LLM's verification. AGSERVE focuses on the reasoning quality compared to speculative decoding rather than the responses' consistency with the large models and supports speculative decoding models in its hierarchy.

**LLM Agents.** ReAct [59] refines LLM's response with multiple rounds of interactions, while LangChain [25] provides an interface for agent coding. Beyond the wide application of LLM agents [33, 45, 29], recent works also propose multi-agent collaboration [49, 32]. Despite its rapid development, we still see most LLM agents rely on a sole model [25, 31]. AGSERVE provides a set of APIs that wrap up the model cascading and quality maintenance process of agent serving.

# 10    Conclusion

We present AGSERVE, the first agent serving system designed to overcome the inherent cost-quality tradeoff in agent serving. AGSERVE addresses the unique demands of agent serving, including predictable request patterns, growing reasoning complexity, and dynamic resource requirements. It combines session-aware KV cache management, quality-aware model cascading, and adaptive resource scheduling to support efficient, high-quality interactions across agent sessions. Experimental results show that AGSERVE significantly reduces latency and resource costs while maintaining service quality on par with state-of-the-art LLM serving systems.

## Acknowledgment

We thank anonymous reviewers for their constructive comments. Li Chen is the corresponding author. This work is supported by the National Natural Science Foundation of China under Grant U21B2022 and the Beijing Outstanding Young Scientist Program (No. JWZQ20240101008)

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

# Appendix

## A    Proof of Theories

### A.1    In-place KV cache calibration

RoPE is a relative positional encoding method and uses circular function to do positional encoding. In formal, we have a rotating matrix relative to the token position $p$,

$$R(p) = \begin{pmatrix} \cos(p\theta_0) & -\sin(p\theta_0) & 0 & \cdots & 0 \\ \sin(p\theta_0) & \cos(p\theta_0) & 0 & \cdots & 0 \\ 0 & 0 & \cos(p\theta_1) & \cdots & 0 \\ \vdots & \vdots & \vdots & \ddots & \vdots \\ 0 & 0 & 0 & \cdots & \cos(p\theta_{N/2-1}) \end{pmatrix}$$

, where $\theta_i$ is a dimension relative to the dimension. The RoPE performs $Q_p = R(p)(x_p W_Q)$, $K_p = R(p)(x_p W_K)$ for the token $x_p$. We denote $\hat{K}_p = x_p W_k$. In other words,

$$K_p = \begin{pmatrix} \cos(p\theta_0) * \hat{K}_{p,0} - \sin(p\theta_0) * \hat{K}_{p,1} \\ \sin(p\theta_0) * \hat{K}_{p,0} + \cos(p\theta_0) * \hat{K}_{p,1} \\ \cos(p\theta_1) * \hat{K}_{p,2} - \sin(p\theta_1) * \hat{K}_{p,3} \\ \vdots \\ \sin(p\theta_{N/2-1}) * \hat{K}_{p,N-2} + \cos(p\theta_{N/2-1}) * \hat{K}_{p,N-1} \end{pmatrix}$$

When truncation happens, we still have $\hat{K}'_{p'} = \hat{K}_p$ which does not change.

$$K'_{p'} = \begin{pmatrix} \cos(p'\theta_0) * \hat{K}_{p,0} - \sin(p'\theta_0) * \hat{K}_{p,1} \\ \sin(p'\theta_0) * \hat{K}_{p,0} + \cos(p'\theta_0) * \hat{K}_{p,1} \\ \cos(p'\theta_1) * \hat{K}_{p,2} - \sin(p'\theta_1) * \hat{K}_{p,3} \\ \vdots \\ \sin(p'\theta_{N/2-1}) * \hat{K}_{p,N-2} + \cos(p'\theta_{N/2-1}) * \hat{K}_{p,N-1} \end{pmatrix}$$

Here, we verify the correctness of Eq. 1.

$$\begin{aligned} K'_{p',2i} &= \cos((p+\delta_p)\theta_i) * \hat{K}_{p,2i} - \sin((p+\delta_p)\theta_i) * \hat{K}_{p,2i+1} \\ &= (\cos(p\theta_i)\cos(\delta_p\theta_i) - \sin(p\theta_i)\sin(\delta_p\theta_i)) * \hat{K}_{p,2i} - \\ &\quad (\sin(p\theta_i)\cos(\delta_p\theta_i) + \cos(p\theta_i)\sin(\delta_p\theta_i)) * \hat{K}_{p,2i+1} \\ &= \cos(\delta_p\theta_i)(\cos(p\theta_i)\hat{K}_{p,2i} - \sin(p\theta_i)\hat{K}_{p,2i+1}) - \\ &\quad \sin(\delta_p\theta_i)(\sin(p\theta_i)\hat{K}_{p,2i} + \cos(p\theta_i)\hat{K}_{p,2i+1}) \\ &= \cos(\delta_p\theta_i)K_{p,2i} - \sin(\delta_p\theta_i)K_{p,2i+1} \\ &= \cos(-\delta_p\theta_i)K_{p,2i} + \sin(-\delta_p\theta_i)K_{p,2i+1} \end{aligned}$$

On the other hand, for $K'_{p',2i+1}$

$$\begin{aligned} K'_{p',2i+1} &= \sin((p+\delta_p)\theta_i) * \hat{K}_{p,2i} + \cos((p+\delta_p)\theta_i) * \hat{K}_{p,2i+1} \\ &= (\sin(p\theta_i)\cos(\delta_p\theta_i) + \cos(p\theta_i)\sin(\delta_p\theta_i)) * \hat{K}_{p,2i} + \\ &\quad (\cos(p\theta_i)\cos(\delta_p\theta_i) - \sin(p\theta_i)\sin(\delta_p\theta_i)) * \hat{K}_{p,2i+1} \\ &= \cos(\delta_p\theta_i)(\sin(p\theta_i)\hat{K}_{p,2i} + \cos(p\theta_i)\hat{K}_{p,2i+1}) + \\ &\quad \sin(\delta_p\theta_i)(\cos(p\theta_i)\hat{K}_{p,2i} - \sin(p\theta_i)\hat{K}_{p,2i+1}) \\ &= \cos(\delta_p\theta_i)K_{p,2i+1} + \sin(\delta_p\theta_i)K_{p,2i} \\ &= \cos(-\delta_p\theta_i)K_{p,2i+1} - \sin(-\delta_p\theta_i)K_{p,2i} \end{aligned}$$

## A.2 The Optimality of ECE Policy under one Setting

In this section, we prove the optimality of SAS's ECE policy in the case that each session cache is of the same size.

Suppose there exists another eviction policy $A$ such that it achieves higher session cache hit rate than ECE on some session access sequence.

Let $H_{\text{ECE}}$ be the hit rate achieved by ECE on session access sequence $S$, and $H_A$ be the hit rate incurred by $A$. By assumption,

$$H_A > H_{\text{ECE}}.$$

Consider the first time $t$ at which $A$ and ECE make different decisions about which session to evict from the cache.

Let the sessions in cache just before time $t$ be the same for both policies, and suppose the current access at time $t$ requires an eviction. ECE evicts session $sc_o$ and $A$ evicts a different session $sc_a \neq sc_o$.

Since ECE is defined to evict the session that will not be used for the longest time in the future, $sc_o$ is the optimal choice given perfect knowledge of future accesses.

There are two cases to consider:

1. If $sc_a$ is used *sooner* than $sc_o$ in the future, then $A$'s decision will cause a cache miss *before* ECE does, decreasing $H_A$.

2. If $sc_a$ is used *later* than $sc_o$ or never used again, then ECE would have chosen $sc_a$ instead, contradicting the definition of ECE.

In either case, $A$ cannot produce a higher cache hit rate than ECE, which contradicts our assumption. Thus, the optimality of SAS's ECE policy is proved in this case.

## B  Additional Details of Background

As the R-Judge observes, an agent may face invalid actions or preset rule violation prior to final success or reaching round limit. We present a detailed analysis of agent outcomes for 20 Alfworld agent cases in Table 4. There is an increasing trend for invalid action or format in the later stages and smaller models.

**Table 4:** Outcome comparison between Llama-8B and Llama-70B.

| Model | Success | Reach Round Limit | Invalid Actions | Rule Violation |
|---|---|---|---|---|
| Llama-8B | 35% | 40% | 15% | 10% |
| Llama-70B | 65% | 30% | 5% | 0% |

## C  Additional Details of Training

The Chatbot-Arena dataset includes 33k pairs of human-labeled preference data of the following models.

- fastchat-t5-3b
- chatglm-6b
- mpt-7b-chat
- vicuna-7b
- stablelm-tuned-alpha-7b
- dolly-v2-12b
- oasst-pythia-12b
- alpaca-13b
- koala-13b
- llama-13b
- vicuna-13b
- vicuna-13b-v1.2
- gpt4all-13b-snoozy
- wizardlm-13b
- RWKV-4-Raven-14B
- guanaco-33b
- palm-2
- claude-instant-v1
- claude-v1
- gpt-3.5-turbo
- gpt-4

### C.1  Q-Judge Training

To facilitate training, we group the models in the arena into three tiers with increasing model sizes and higher ranking. A question is classified as solvable by the smaller model if the smaller one either

**Table 5:** Data Distribution of Q-Judge Training Set

|      | Small   | Medium  | Large   | Accuracy |
|------|---------|---------|---------|----------|
| GT   | 53.85%  | 33.52%  | 12.63%  | N/A      |
| Eval | 44.35%  | 51.55%  | 4.10%   | 52.45%   |

**Table 6:** Data Distribution of R-Judge Training Set

|      | 0 (Unsatisfactory) | 1 (Satisfactory) | Accuracy |
|------|--------------------|------------------|----------|
| GT   | 49.59%             | 50.41%           | N/A      |
| Eval | 43.10%             | 56.90%           | 82.76%   |

wins or ties against a larger model's response; otherwise, it is considered challenging enough to require a more powerful model.

We train the Q-Judge for 10 epochs on top of BERT [22] with a batch size of 16 and a warm-up step of 500, utilizing one A6000 GPU. We choose BERT as the backbone since Q-Judge is less frequently called and the routing task requires high precision. The training takes 2.9 hours. Under the customized loss, the Q-Judge achieves 52% accuracy on the evaluation set. Please be noted that the definition of accuracy here is different from the one in §7.4.2, where we define a much looser restriction due to the difficulty of annotation.

The distribution of the Q-Judge train set and the results on the evaluation set are shown in Table 5. *GT* shows the distribution of ground truth labels and *Eval* shows the distribution of Q-Judge generated labels on the evaluation set.

### C.2   R-Judge Training

We train the R-Judge on top of DistilBERT [43] using another customization of the Chatbot-Arena dataset. We choose DistilBERT as the backbone since R-Judge is more frequently called, and its efficiency is extremely important. We label the winner's response as capable, while the loser's as incapable. For a tie match, we only take the smaller model's response as capable and drop the larger LLM's. We train the R-Judge for 10 epochs with a weight decay of 1e-2 and warm-up steps of 500. The training takes 17 minutes. $RJ_{0.5}$ achieves an 82.8% agreement rate with human-selected preferences on the evaluation set.

The distribution of the R-Judge train set and the result on the evaluation set are shown in Table 6 with GT representing ground truth.

## D   Additional Implementation Details of AGSERVE

We implement the AGSERVE system in approximately thousands lines of Python and CUDA code. AGSERVE provides an interface for automatic agent programming and supports model hierarchy customization with agile agent serving support.

**LLM Serving.** The SAS part of AGSERVE is based on the SoTA LLM serving system, vLLM [24]. AGSERVE supports session cache manager and two-level load balancing on the instances to accelerate inference speed. We extend the APIs to record the session ID and session cache reuse pattern information (for truncation and restoration) for each request. We also provide more APIs to support the dynamic model allocation and proactive session cache release. AGSERVE supports starting session-aware instances of preset models via Ray [35].

**Algorithm in ECE Eviction Decision.** The pseudo code of the algorithm used in eviction decision is shown in Alg. 1, as we describe in §4. In L1 to L15, we adopt a knapsack-like algorithm to give the optimal recomputation overhead with the number of actual released blocks and evicted sessions. In L18 to L19, the algorithm calculates the recomputing penalty for each session. In L20 to L26, the algorithm walks through different $i$s, which defines the actual number of sessions to evict. AGSERVE predicts ETA in the following manner. First, AGSERVE maintains the metadata of each session. When a session starts, AGSERVE initializes the metadata by the agent type. Throughout the session, AGSERVE updates the metadata with the last several rounds' round duration, and last arrival time data. AGSERVE will use the average round duration to calculate the ETA. The ETA grows if the next request has not arrived at the original ETA time, and finally reaches infinite in case of unexpected session interruption. The ETA prediction will not be drastically affected by the unexpected behaviors, since it keeps track of multiple rounds of history data. It is robust to workload shift, since each session is treated independently.

**Algorithm 1:** ECE Eviction Policy

---

**Input:** Number of ongoing sessions $n$, number of running sessions $r$, number of waiting sessions $w$, session cache size (can be zero) $l[1\ldots n]$ ranked by ETA, maximum batch size $K$, prefilling per-block overhead $P$, decoding time consumption $D$

**Output:** Available session nums $k$, sessions to evict $E$

**1** **Function** `Knapsack`($i$, $W$):

**2**     Initialize $dp[0] \leftarrow 0, dp[1\ldots W] \leftarrow \infty, S[0..W] \leftarrow []$;

**3**     Initialize $asp \leftarrow 0, aval \leftarrow \infty, as \leftarrow []$;

    `// The first i sessions cache shall remain in the cache and is not considered in eviction`

**4**     **for** $x \leftarrow i+1$ **to** $n$ **do**

**5**         **for** $j \leftarrow W$ ***downto*** $W - l[x]$ **do**

            `// for the cases where released blocks exceeds the requiring W`

**6**             $val = dp[j] + v[x]$;

**7**             **if** $val < aval$ **then**

**8**                 $aval \leftarrow val$;

**9**                 $as = S[j] + [x]$;

**10**                 $asp = j + l[x]$;

                `// No need to update dp, since no better solution comes based on asp`

**11**         **for** $j \leftarrow W - l[x]$ ***downto*** $0$ **do**

            `// for the cases where released blocks does not exceed the requiring W`

**12**             $val = dp[j] + v[x]$;

**13**             **if** $val < dp[j + l[x]]$ **then**

**14**                 $dp[j + l[x]] \leftarrow val$;

**15**                 $S[j + l[x]] = S[j] + [x]$;

**16**     **return** $aval, asp, as$;

**17** $k \leftarrow D_r \cdot (n - r), E \leftarrow []$;

**18** $rsp \leftarrow 0, asp \leftarrow 0$;

`// Calculate the recomputation penalty for each session`

**19** **for** $i \leftarrow 0$ **to** $n$ **do**

**20**     $v[i] \leftarrow P \cdot l[i] \cdot (n - i + 1)$;

`// Walk through all possible batch size.`

**21** **for** $i \leftarrow 1$ **to** $min(w, K - r)$ **do**

**22**     $rsp \leftarrow rsp + rs[i] - l[i]$;

    `// Check if new session needs to be evicted, improve efficiency`

**23**     **if** $asp < rsp$ **then**

**24**         $t, asp, E_i \leftarrow$ `Knapsack`($i$, $rsp$);

**25**     $val = t + D_{r+i} \cdot (n - r - i)$;

**26**     **if** $val < k$ **then**

**27**         $k \leftarrow i, E \leftarrow E_i$;

**28** **return** $k, E$

---

Each SAS runs its ECE algorithm independently. The theoretical algorithm complexity is $O(k*n*W)$. We optimize the large constant W (knapsack size) with a Python dictionary in our implementation. Due to the long context of each session, the number of session cache maintained in the context is limited. As a result, even in the worst case, the latency per ECE decision remains under 0.1 milliseconds, which we find negligible compared to typical inference latency.

**Model Hierarchy.** AGSERVE adopts a three-layer hierarchical model structure. We choose the 8B (Llama-8B) and 70B (Llama-70B) versions of the widely adopted Llama-3 [14] as the first two layers. These two models also outperform in multiple benchmarks [8, 2]. We adopt GPT-4o [1] at the third layer as an external API, which is one of the best models in multiple agent benchmarks [29, 51].

**Table 7:** AGSERVE APIs

| API | Description |
|---|---|
| **Implementation Building** | |
| set_sp() | Set the shared prefix for the implementation. |
| set_tsp() | Set the task specific template for the implementation. |
| set_tools() | Set the tools and its descriptions for the implementation. |
| **Task Executing** | |
| set_models() | Designate the cascading models, ranked by the order from weak to strong. |
| set_tau() | Designate the quality threshold for $RP(\cdot)$ |
| set_nu() | Designate the response checking frequency for $RP(\cdot)$ |
| set_task() | Set the task and assemble the prompt. |
| adjust_tool() | Adjust the availability of tools, and add/remove tools at the present round. |
| inject() | Inject the observation. |
| action() | Query the LLM with the existing prompt, and return the LLM-generated actions. |
| req_change() | Request a general migration. |
| req_update() | Request a service-upgrade migration. |
| go_back() | Restore the saved status. |

We choose the three-layer hierarchy due to the following reasons. Firstly, the physical limits of our testbed constrain the number of LLMs we can deploy. Secondly, we ensured that the models at each layer exhibit a substantial performance gap in reasoning quality, which takes advantage of model diversity. The benefit of Llama also applies to other models adopting the Llama structure, including the Qwen family [57]. Notably, **AGSERVE does not benefit from dictionary reuse**, since the tokenization process only takes no more than 2 milliseconds even under the full context window, and is not comparable to the TTFT. As a matter of fact, AGSERVE re-tokenize the prompt for each request. The cascading structure in AGSERVE is flexible. Adding or removing layers is straightforward: users only need to customize the Chatbot-Arena dataset and retrain the QMM. Thirdly and importantly , both the Llama-8B and Llama-70B models are adjustable models, as stated in §6, and are widely adopted in agents. They are from the same family and share the same tokens and dictionary. GPT-4o is selected due to its outstanding performance in multiple benchmarks, demonstrating AGSERVE's support for commercial model APIs besides SAS.

**Fault Tolerance.** AGSERVE is capable to handle the possible failures in distributed inference. If an instance fails, the client sends a request to the scheduler for rerouting. If the scheduler fails, AGSERVE defaults to a regular LLM serving system with a session cache enabled, establishing sessions between the agent and a random instance.

**Agent Programming.** AGSERVE provides a set of APIs like LangChain [25] for agent programming. Agent operators can build their own agent implementation with available tools and prompt templates The clients can also designate the used LLMs, the quality threshold, *etc.* through the APIs. We list the APIs for programmers to build the agent implementations as well as those for clients to use AGSERVE in Table 7. The agent operates on the `Session` class to perform LLM query and quality maintenance. Each agent can pass the $\tau, ip, p$ parameters and the quality threshold $\theta$ to `Session`. For each query, the `Session` class will return a quality indicator in addition to the response. For the restorable agents, the quality indicator may instruct them to save or restore the previous history.

## E  Additional Details of Evaluations

### E.1  Implemented Agents

We implement the following four agent implementations to demonstrate the performance of AGSERVE.

- The AlfWorld (AW) [45] is an embodied household assistant navigating and interacting within simulated environments. AW is a standard agent that includes multiple rounds of interaction to attempt and fail.

- The card game (CG) [29] plays a two-player game called AquaWar against a random-acted baseline. The client performs two types of actions: attack and guess. Prompt templates of the two types vary. We implement CG with two AGSERVE sessions using multi-agent collaboration. The two sessions perform model cascading independently.
- The knowledge graph (KG) [29] queries knowledge graphs to answer a question. We start a Freebase knowledge graph via Virtuoso [4, 15] on a CPU node with 256 GB memory. Queries do not alter KG, so we implement KG as a restorable agent introduced in §5.2.
- The Mind2Web (M2W) [12] navigates and performs tasks across various domains of real-world websites. While it presents a complex challenge, an LLM usually completes the task within a few interaction rounds.

### E.2 Quality for Different Agents

We outline the computation of the quality metric in the section. $sc$ is set to 25 if the session ends successfully in status `COMPLETED` or `TASK_LIMIT_REACHED`; otherwise, it is set to 0.

#### E.2.1 AW Agent Quality

For the AW agent, the quality calculation is as follows:

$$Q_{AW} = sc + vc + \left(25 - \frac{\text{rounds}}{2}\right)$$

The base quality is set to 25, with an additional 50 points of $vc$ if there is a valid result. If the agent accomplishes the task, $vc$ is set to 50, otherwise, 0. While fewer interaction rounds mean fewer moves for the embodied agent, a score of 25 is added with a penalty of 0.5 for each move.

#### E.2.2 CG Agent Quality

For the CG agent, the quality is calculated based on meta-performance metrics:

$$Q_{CG} = \frac{\text{full\_play}}{\text{test\_times}} \times 25 + \frac{\text{harm\_on\_enemy}}{\text{total\_HP} \times \text{test\_times}} \times 50$$
$$+ 25 \times \max(\frac{\text{round\_num}}{50 \times \text{test\_times}}, 1)$$

The quality of CG partly breaks the common requirement of quality factors due to its "game" character. It comprises three parts: the normal behavior part by the ratio of full play rounds (in how many test times the CG agent operates normally), the accomplishing part by the win round ratio, and a survival part replacing the efficiency part. In most cases, we find LLM loses the battle. In a losing game, a longer survival should be given a bonus rather than a quick death.

#### E.2.3 KG Agent Quality

For the KG agent, the quality is computed as:

$$Q_{KG} = sc + 50 \times \text{f1} + \left(25 - \frac{\text{rounds}}{2}\right)$$

A base quality of $sc$ is added, followed by the F1 score. The F1 score reflects how well the answer given by the KG agent matches the ground truth. If the F1 score is greater than zero, an additional score of 25 is applied with a penalty based on the number of actions, with a reason similar to the AW.

#### E.2.4 M2W Agent Quality

For the M2W agent, the quality is derived from multiple metrics:

$$Q_{M2W} = sc + \text{element\_acc} \times 25 + \text{step\_sr} \times 25$$
$$+ \text{action\_f1} \times 25$$

This calculation is based on three key metrics: element accuracy, step success rate, and action F1 score. Each metric contributes equally to the overall quality, with a maximum of 25 points each.

### E.3 Cost of Open-Source Models

We use the retail price of cloud server rentals to estimate the cost of open source models. To our knowledge, the cost of open-source models could be **significantly lower** (30% to 60% lower) for

**Table 8:** Cost of AGSERVE and baselines for Alfworld under different pricing plans.

|  | AGSERVE | Llama-8B | Llama-70B | GPT-4o | RouteLLM | Cascade |
|---|---|---|---|---|---|---|
| Original Cost | 0.0177 | 0.0023 | 0.0376 | 0.1073 | 0.0337 | 0.1395 |
| Cost(-30% Rent) | 0.0137 | 0.0016 | 0.0263 | 0.1073 | 0.0324 | 0.1353 |
| Cost(-60% Rent) | 0.0097 | 0.0009 | 0.0151 | 0.1073 | 0.0312 | 0.1311 |
| Quality | 90 | 21 | 72 | 94 | 25 | 69 |

self-operated data centers or long-term users with Azure's spot or reserved plan[2]. The lower cost could further exhibit the advantage of AGSERVE against the GPT-4o-only approach.

### E.3.1 The A6000 Testbed

We find the instance of the most similar computing capability offered by cloud service providers to our testbed is `ecs.gn7i-8x.16xlarge` on Alibaba Cloud with 8 A10 GPUs. The rent for this instance is approximately $10 per hour.

The Llama-8B model requires only 1/4 of the node and can run 32 sessions concurrently with no throughput loss. The price of Llama-8B per session is $10 \div 3600 \div 4 \div 32 = \$(2.17 \times 10^{-5})/s$.

The Llama-70B model requires one node and can run 16 requests concurrently with no throughput loss. The price of Llama-70B per session is $10 \div 3600 \div 16 = \$(1.73 \times 10^{-4})/s$.

AGSERVE requires 5/4 of the node to perform both 8B and 70B models. Though with higher throughput, we calculate the price of it as $\frac{5}{4} \times \$1.73 \times 10^{-4} = \$(2.17 \times 10^{-4})/s$.

### E.3.2 The A800 Testbed

We find the instance of the most similar computing capability offered by cloud service providers to our A800 testbed is `ND96amsr A100 v4` offered by Azure. The rent for this instance is approximately $33 per hour, with tax included.

All three agent serving systems evaluated in §7.3 utilize two $8 \times 80$GB A800 instances. We find out that the testbed is capable of handling about 64 sessions (requests) simultaneously without computing resource preemption. We define the price of using the A800 testbed for each session as $33 \times 2 \div 3600 \div 64 = \$(2.86 \times 10^{-4})/s$.

### E.4 AGSERVE Settings

We set the quality control threshold $\theta$ of $RJ_\theta(\cdot)$ to 0.5. The reasoning quality check frequency $\nu$ is set to 4 for all agents except M2W. AGSERVE checks the LLM's reasoning quality for every response ($\nu = 1$), as an M2W session usually finishes in 2 to 3 rounds.

### E.5 Details of End-to-End Cascaded Serving

In §7.2, we show that AGSERVE shows superior performance under the retail rental price. In this section, we display the cost of AGSERVE and other baselines for operating the Alfworld agent under different pricing plans in Table 8. AGSERVE's rental price is still lower than that of Llama-70B which has worse quality than it. With a discount in the rental prices, AGSERVE further breaks the current cost-quality tradeoff curve with only 9.04% cost to achieve similar performance with GPT-4o.

### E.6 Details of Multi-Agent Traces

Uniform, Gamma, Poisson, and Burst traces that we evaluated in §7.3 comprise 100 agents each. The uniform, Gamma, and Poisson traces share an average sending rate of 4 seconds per agent. Running traces is very expensive. It costs us more than 500 USD to produce Fig 9 just for the OpenAI API.

### E.7 Details of Ablation Studies

### E.7.1 SAS Efficiency

We set the `max_memory_utilization` of SAS to 0.37 (equivalent to 17.8GB, 3GB of CUDA graph not included) and the `max_model_len` to 4096 on one A6000 GPU. Under this setting, there are 745 blocks for KV cache, fitting 6 to 3 sessions depending on their round. This setting promotes eviction and truncation during agent execution to demonstrate the efficiency of SAS's ability better. Nevertheless, this setting could be very common in real-world serving scenarios [23], where GPUs have limited memory, such as NVIDIA(R) RTX 4090 (24GB) and A10 (24GB) GPUs.

---

[2]See `https://azure.microsoft.com/en-us/pricing/details/virtual-machines/linux/`.

**Table 9:** QMM finishes evaluation job in real-time.

|              | Q-Judge | R-Judge | AW  | CG   | KG      | M2W  |
| ------------ | ------- | ------- | --- | ---- | ------- | ---- |
| Overhead (s) | 0.05    | 0.03    | 0.3 | 0.04 | 0.005-2 | 0.03 |

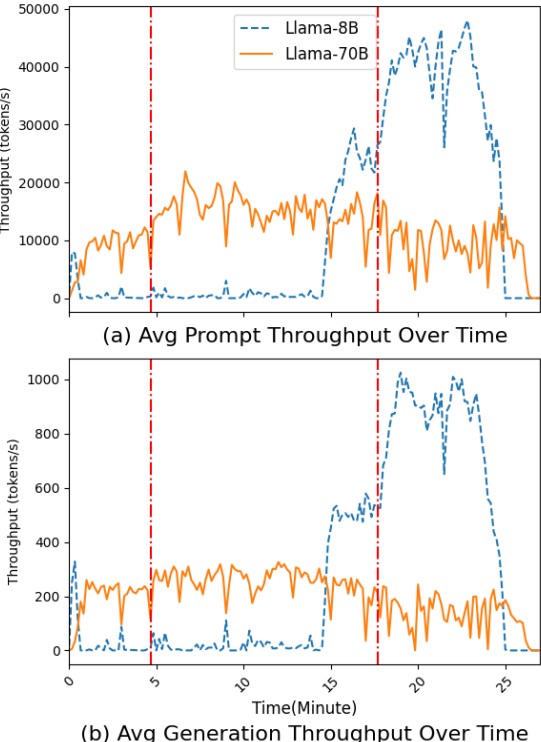

(a) Avg Prompt Throughput Over Time

(b) Avg Generation Throughput Over Time

**Figure 15:** The prefilling and generation throughput of different models in AGSERVE. The thin red line at 4.67 min shows the launch of a two-A800 70B instance and a half-cut of a four-A800 8B instance. The bold red line at 17.67 min shows the two-A800 70B instance replaced by an 8B instance.

### E.7.2 QMM Efficiency

To minimize the bias, we only show the observation and LLM's response round by round to our annotators, with no previous knowledge of the session's final result or the model size. We also utilize the voting mechanism and each label is agreed by at least three annotators.

We compare the overhead of running Q-Judge, R-Judge, and the response time for four agents in Table 9. The results show that the R-Judge can finish in the time of response generation in most cases. The Q-Judge only introduces negligible overhead in its infrequent calling. Please note that we take the time from the generation of request to the generation of first token as TTFT in Fig 14, including the overhead for node routing, which is not counted in other experiments.

### E.7.3 Details and Case Study of Dynamic Allocation

To create an imbalanced demand-supply ratio, we continuously launch agent samples every 5 seconds with $\theta$ at a low value of 0.01, deferring sessions to the Llama-70B instances, mimicking the burst of difficult tasks at the very beginning. After a period, we raise $\theta$ to mimic a change to low-cost pursuit. We forbid AGSERVE from visiting OpenAI APIs to prevent the assistance of external machines.

In this section, we give a case study on the impact of dynamic allocation. The agent launcher keeps launching the agent at a given speed and shifts the quality maintenance threshold at some time point. Fig 15 depicts the prompt and generation throughput of different models during the experiment. The prompt throughput shows the prefilling speed, while the generation throughput reflects the decoding speed. AGSERVE first moves 2 GPUs from the 8B instance to a 70B instance due to the high volume of 70B instances. And it dynamically moves the 2 GPUs back when the demand for 8B models

| Method | Cost (.01$) | Quality | Overhead(s/req) | Gross Cost(.01$) |
|---|---|---|---|---|
| Human | **1.505** | **91.75** | 14.71 | 2.944 |
| AGSERVE | 1.768 | 89.57 | - | **1.768** |

**Table 10:** Evaluations of human-like judges in agent serving of Alfworld. The overhead column shows the time it takes for a human to judge the satisfaction per request. Gross cost shows the cost if the human-judge overhead is included in the calculation.

**Table 11:** Comparison of centralized (C) and distributed(D) scheduling by e2e latency (EL), quality (Q), and scheduling latency (SL) under different agent launch rates (LR). EL and SL are in seconds.

| LR (s/agent) | EL-C | Q-C | SL-C | EL-D | Q-D | SL-D |
|---|---|---|---|---|---|---|
| 0.8 | 64.76 | 84.18 | 0.42 | 47.66 | 85.83 | 0.09 |
| 0.4 | 97.45 | 83.39 | 0.43 | 60.12 | 85.86 | 0.08 |
| 0.2 | 129.99 | 85.03 | 0.46 | 68.94 | 85.06 | 0.09 |

increases. Notably, AGSERVE keeps the functionality of both models throughout the time, since there are always requirements for both models. As we discussed in §7.4.3, we face 6% failures in FS1, FS2 and AGSERVE due to the LLM's limited capability in long-context scenarios. In routine operation, AGSERVE will choose to upgrade the model to GPT-4o.

### E.8 Performance of an Ideal Human-Like Judge

In a fast-paced request pattern in agent serving, it is impossible for a human to judge in real-time. Our experience shows that it takes 10 seconds or more for humans to read and understand the context before rating the response, missing the latency objective of many agents. Fig 13 depicts a 75.8% accuracy in response quality judgment relative to human preference. The KV cache of different models can not be reused. AGSERVE adopt periodical quality check and retry strategy as stated in §5.2 to avoid excessive model migration, which may lead to extra overhead of KV cache re-computation. In this experiment, we want to show the performance of an ideal human-like judge on the AW agent. To do this, we let a human expert determine whether to migrate or not for each response, and evaluate the cost and quality of each agent serving. We do **not** include the time for humans to judge in the cost and latency calculation. Table 10 shows the performance of the human judge, which only outperforms AGSERVE with margin benefits, showing that R-Judge is strong enough for the three-layer session lifetime management. We acknowledge that it may not apply to cascades with more layers of models.

## F AGSERVE on a Large Scale

While AGSERVE employs a centralized scheduler for distributed agent serving, it is designed to support distributed scheduling, where each district has its own RS. We simulate AGSERVE's performance on a large scale with the following approaches:

1. The inference latency on instances is based on a sleep operation, and we profile the inference duration by the GPU setting, concurrent requests, and model size.
2. The quality issues are triggered at a certain probability related to the size/capability of LLM.
3. We define a virtual round-to-go. One interaction round with a larger LLM has a higher probability of decreasing the virtual round-to-go more than a smaller one.

We simulate the behavior of a 60-node cluster with 8 GPUs per node. The result is shown in Table 11, showing that the distributed scheduling version of AGSERVE reduces e2e latency while accelerating scheduling speed and improving fairness in the cluster.

