# OpenReview forum: "Transcending Cost-Quality Tradeoff in Agent Serving via Session-Awareness"
_NeurIPS.cc/2025/Conference — NeurIPS 2025 poster_

### Official Review · Reviewer_E9yH · 2025-06-22

**Clarity:** 3
**Significance:** 3
**Originality:** 3
**Rating:** 4
**Confidence:** 2

**Summary:**

This paper presents AGSERVE, a session‐aware agent serving system designed to “transcend” the traditional cost–quality tradeoff in LLM‐based agent workflows. By combining (1) in‐place RoPE calibration and an ETA‐based cache eviction policy to boost KV‐cache reuse under middle‐truncation, (2) a Quality Maintenance Module (QMM) with a Q-Judge for initial model selection and an R-Judge for runtime quality monitoring and migrations, and (3) a dynamic GPU scheduler for demand‐driven resource allocation, AGSERVE claims up to 83.5% cost savings versus GPT-4o at equivalent quality, 1.8× quality gain at fixed cost, 2.86× cache‐hit improvement, and 1.2× end-to-end speedup on real multi-GPU testbeds across four agent benchmarks.

**Questions:**

1. Have you evaluated AGSERVE on any non-programming agent workloads to assess generality?

2. Can you provide false-positive/false-negative curves for the R-Judge, and quantify migration overhead when misclassifications occur?

3. What is the worst-case decision latency of the ECE algorithm as agent count and cache size scale?

**Ethical Concerns:**

["NO or VERY MINOR ethics concerns only"]

**Final Justification:**

I assign a final score of 4 (Borderline Accept). The clarifications and new experiments provided during the rebuttal largely resolve my concerns. Please incorporate these experiments and clarifications into the next revision of the manuscript.

**Limitations:**

Experiments are confined to four programming‐agent tasks, leaving AGSERVE’s effectiveness on dialogue, planning, or other agent workloads unverified.

**Quality:**

3

**Strengths And Weaknesses:**

### Strengths
1. Tailors cache and eviction policies to multi-round agent contexts, yielding substantial cache reuse and latency gains.

2. Integrates caching, quality judgers, and resource scheduling under a coherent API, simplifying adoption.

3. Thorough ablations isolate the benefits of in-place calibration, ECE vs. LRU eviction, Q-Judge/R-Judge overheads, and dynamic allocation across two testbeds (A6000 and A8000 clusters).


Weaknesses
1. All experiments target four programming-agent tasks on AgentBench. It remains unclear whether AGSERVE’s session-aware optimizations generalize to non-coding agents (e.g., dialogue, planning).

2. While Q-Judge and R-Judge training details and accuracy/distribution tables are provided, the paper omits any analysis of misclassification trade-offs (false-positive vs. false-negative rates) or calibration curves that would clarify migration costs under varying thresholds.

3. The ECE eviction decision algorithm is described in pseudocode, but its runtime complexity and decision-latency overhead under tight memory budgets are neither analyzed nor experimentally measured.

---

> ### Author Rebuttal · Authors · 2025-07-29
>
> Thank you for your detailed review and feedback.
>
> [Weakness 1, Question 1, Limitations 1] Have you evaluated AGSERVE on any non-programming agent workloads to assess generality?
> > Thank you for your question. We implement four agents, AlfWrold (simulated household jobs / embodied agents), Card Game (card games/ game playing), Knowledge Graph(knowledge graph / Q&A dialogue) , Mind2Web (web browsing / planning) on top of AgServe, spanning across a wide range of agent application and request patterns. As shown in Figure 8, AgServe consistently outperforms baselines across all four domains by pushing the cost-quality frontier. We will clarify this generality and add a forward reference to Appendix D.1 at the beginning of Section 7 in the revision.
>
> [Weakness 2, Question 2] Can you provide false-positive/false-negative curves for the R-Judge, and quantify migration overhead when misclassifications occur?
> > Thanks for your question. Sure! Here is some data points on the False-Positive/False-Negative curve. We’ll add the curve to our revision. False positive means that the R-Judge mistakenly classifies a satisfactory result as unsatisfactory, while false negative menas that the R-Judge takes an unsatisfactory one in the ground truth as satisfactory. To reduce the early-migration cost of false positive, AgServe adopts the retry strategy and assesses the response again prior to migration, as described in Section 5.2.
>
> | Data Point | 0     | 1      | 2      | 3      | 4      | 5      |
> |---------------|-------|--------|--------|--------|--------|--------|
> | False Positive Rate| 0.00% | 28.09% | 40.06% | 41.22% | 47.93% | 57.55% |
> | False Negative Rate| 33.64%| 17.60% | 8.98% | 3.94%  | 0.91%  | 0.00%  |
>
> > We also conduct a study on the cost of early migration in Appendix D.7. An ideal human judge with no overhead and perfect accuracy can further reduce the cost by 14%, yet only improves the performance by 2.4%. However, human-as-a-judge introduce second-level latency in reality and raises the cost significantly.
>
> [Weakness 3, Question 3] What is the worst-case decision latency of the ECE algorithm as agent count and cache size scale?
> > Thanks for pointing this out. Each SAS runs its ECE algorithm independently. The theoretical algorithm complexity is O(k\*n\*W). We optimize the large constant W (knapsack size)  with a Python dictionary in our implementation. Due to the long context of each session, the number of session cache maintained in the context is limited. As a result, even in worst-case conditions, the observed latency per ECE decision remains under 0.1 milliseconds, which we find negligible compared to typical inference latency. We will clarify this in the final version and include this worst-case estimate.

---

> ### Comment · Reviewer_E9yH · 2025-08-04
>
> I appreciate the clarification and new experiments. I have no further comments.

---

### Official Review · Reviewer_j2me · 2025-07-01

**Clarity:** 1
**Significance:** 2
**Originality:** 3
**Rating:** 4
**Confidence:** 3

**Summary:**

The work proposes an agent serving system called AgServe, which jointly optimize KV cache reuse, model selection and resource allocation, reducing the agen serving cose while achieving higher quality results.

The author starts by arguing that the current serving systems are largely developed for serving generic LLMs and are not optimized for the agent workflows. The authors have identified the following agent serving characteristics:
- predictable and high frequency interaction: the interactions with agents are light-weight (less tokens) and frequent
- growing context and increasing difficulties: agents interact in multi-round fashion, making the context length grows rapidly
- unique prompt formatting: agent has specific ways of formatting prompts

The main idea is to predict sessions to reduce inference latency and depending on the phase of the session, use different-sized models to reduce cost.

The major challenges to achieve this are:
session aware cache policy: current system’s reuse and eviction policies are not well-suited for agent serving
- quality assessment: existing systems do not estimate the session lifetime and do not have the ability to dynamically assess model response quality and thus cannot choose the best model for each stage
- resource allocation: the gpu usage of model switching is not optimized

**Questions:**

- I am curious what the Q-Judge / R-Judge are practice? Are they also LLMs? How can you ensure these instances fulfill the designated tasks? E.g., the Q-Judge should select the most cost-efficient model for the session by assessing the task’s difficulty, but how can you be sure it chose the most cost-efficient model? Do you need to train them or provide additional context? Similarly, the R-Judge should evaluate the reasoning quality, how do decide the consistency in evaluation and how do you set the standard / threshold?
- The author mentioned the customized CUDA kernels in chapter 7, what exactly does the kernels do and why do you need them? There is no explanation of that.
- The ECE eviction policy seems reasonable, but it comes with additionally computation overhead, because it needs to make prediction? LRU follows the simple but universal rule of least recently used to evict cache and since LRU has been adopted by the two most popular inference framework vLLM and SGLang, I believe it has shown robustness in large scale caching. Whether ECE is truely beneficial needs more large scaled validations

**Ethical Concerns:**

["NO or VERY MINOR ethics concerns only"]

**Final Justification:**

The authors have addressed most of my comments and promised to update the paper accordingly with more explanations. I feel like this paper can have a borderline accept, but still needs some polish in the writing and presentation to qualify for better score.

**Limitations:**

- Generally I feel like the delivery of this paper needs further improvement to convey their message more cleaner; currently they keeps on mentioning new concepts, without previous definition or explanation, making the reading difficult. For example, in the beginning of section 7, the author mentioned different agens (AW, CG, KG, M2W), but you have to search through the whole paper and find the definition in the Appendix, or in the ECE Eviction Policy section, it states the formula for calculating the eviction cost, and the P is also not defined. There are other examples like this.
- The strictness of R-Judge theta seems to be an important hyperparamter, as it greatly controls the decision of R-Judge, but how it is obtained and quantified seems unclear (judging from the section 5.2 and the Appendix B)
- Model allocation seems to be too simplified, the RS profiles the nunimum number of GPUs required, but in practise, increasing the number of GPUs will lead to faster inference, this trade-off should be considered when allocating memory

**Paper Formatting Concerns:**

I did not notice any significant formatting issues.

**Quality:**

2

**Strengths And Weaknesses:**

Strengths

- clear motivation and detailed problem analysis: the authors start by claiming that there does not exist a serving system that is optimized for serving LLM agents, then
analyzing the problems of using the generic LLM’s serving systems, making its motivation clear


Weaknesses

- The idea of introducing a Q-Judge and R-Judge is interesting, but I am having doubts that the mentioned training of Q-Judge / R-Judge is sufficient to accomplish their designed roles, by only training on Chatbot-Arena for such little time, also the only training on this dataset seems limited, leaving doubt that whether the trained Q-Judge and R-Judge is able to handle all kinds of tasks, or it would require task specific training

---

> ### Author Rebuttal · Authors · 2025-07-29
>
> Thanks for your feedback, which helps us improve the quality of our manuscript.
>
> [Weakness 1, Question 1] （1）I am having doubts that the mentioned training of Q-Judge / R-Judge is sufficient to accomplish their designed roles, by only training on Chatbot-Arena for such little time, also the only training on this dataset seems limited, leaving doubt that whether the trained Q-Judge and R-Judge is able to handle all kinds of tasks, or it would require task specific training. （2） I am curious what the Q-Judge / R-Judge are practice? Are they also LLMs? How can you ensure these instances fulfill the designated tasks? E.g., the Q-Judge should select the most cost-efficient model for the session by assessing the task’s difficulty, but how can you be sure it chose the most cost-efficient model? Do you need to train them or provide additional context? Similarly, the R-Judge should evaluate the reasoning quality, how do decide the consistency in evaluation and how do you set the standard / threshold?
>
> > Thanks for your insightful question. We would like to address your concerns twofold.
> > 1. Q-Judge and R-Judge are lightweight classifiers built upon BERT and DistilBERT, respectively. Both judges are trained on customized Chatbot-Arena datasets. Q-Judge learns from RouteLLM [35], which has demonstrated robust performance across a wide range of tasks in routing scenarios without task-specific tuning. To minimize unnecessary use of powerful models ("overkill"), Q-Judge employs a novel cost-aware loss that encourages routing to cheaper models when appropriate. R-Judge focuses on evaluating reasoning quality by detecting whether the response addresses the original task, avoids redundant reasoning steps, and maintains logical coherence.
> > 2. We agree that generalization is an important concern. In Section 7.4.2, we present a micro-benchmark evaluating the two judges across four implemented agents. Q-Judge agrees with human preferences in over 70% of cases, and R-Judge achieves 90.5% recall on detecting low-quality reasoning. These results suggest that despite being trained on a single dataset, the judges generalize well in practice. We will clarify this further in the revision.
>
> [Question 2] The author mentioned the customized CUDA kernels in chapter 7, what exactly does the kernels do and why do you need them? There is no explanation of that.
> > Sorry for the confusion. The customized CUDA kernel is designed to batch tokens from multiple sessions and perform in-place calibration of positional embeddings, as described in Section 4. This operation allows us to better utilize GPU memory and reduce kernel launch overhead in high-throughput settings. We will add a detailed explanation of the kernel’s role and its benefits in the revision.
>
> [Question 3] The ECE eviction policy seems reasonable, but it comes with additionally computation overhead, because it needs to make prediction?  I believe LRU has shown robustness in large scale caching. Whether ECE is truly beneficial needs more large scaled validations.
> > Thanks for your question. We acknowledge the concern about potential overhead. In practice, the ECE policy incurs only about 0.1 milliseconds per eviction decision, which we find negligible compared to overall serving latency of 2 seconds. To achieve this, AgServe maintains lightweight metadata for each session (such as prior arrival times and interval) and leverages statistical estimation to predict the expected time of next arrival (ETA).
> > While LRU is very useful in general LLM serving and widely used by many frameworks, it may evict the session cache just before reuse in agent serving, as depicted in Figure 7. To tackle this problem, we introduce estimated-time-of-arrival to agent serving for the first time and proposes ECE.
> > ECE mainly focus on the KV cache of tokens that are not shared by other sessions (i.e. task-specific template and truncatable). The eviction decision of such content is handled entirely within each node. Therefore, the scale of nodes will not reduce ECE’s effectiveness.  We will include this clarification and the runtime overhead measurement in the revised manuscript.
>
> [Limitations 1] Currently they keeps on mentioning new concepts, without previous definition or explanation, making the reading difficult. For example, in the beginning of section 7, the author mentioned different agens (AW, CG, KG, M2W), but you have to search through the whole paper and find the definition in the Appendix, or in the ECE Eviction Policy section, it states the formula for calculating the eviction cost, and the P is also not defined. There are other examples like this.
> > Thank you for pointing this out. We will revise the manuscript to ensure all technical terms and abbreviations are clearly defined when first introduced. Specifically, we will add a forward reference to Appendix D.1 in Section 7 for the agent definitions (AW, CG, KG, M2W). Additionally, we will clarify that P in the eviction cost formula denotes the penalty of recomputation—i.e., the ratio between prefill and decode latency per token. A thorough pass will be conducted to ensure that all such symbols are properly introduced and explained.
>
> [Limitations 2] The strictness of R-Judge theta seems to be an important hyperparamter, as it greatly controls the decision of R-Judge, but how it is obtained and quantified seems unclear.
> > We appreciate your observation. The strictness parameter $\theta$ in R-Judge controls the pass threshold for response quality. A value of 0 means no response will pass the quality check, while 1 accepts all responses. We train the Q-Judge for strictness of 0.5. Users can adjust this threshold based on their own quality requirements or workload characteristics. We will include an explanation of this hyperparameter and its practical implications in the revision.
>
> [Limitations 3] Model allocation seems to be too simplified, the RS profiles the nunimum number of GPUs required, but in practise, increasing the number of GPUs will lead to faster inference, this trade-off should be considered when allocating memory
> > Thanks for your insight. While RS profiles the minimum number of GPUs required to fit a model, it does not restrict allocation to the minimum. AgServe supports tensor parallelism, and RS can assign more GPUs than the minimum when available. For example, if one GPU suffices but six GPUs are idle, RS will spawn two SAS instances using 4 and 2 GPUs respectively, allowing faster inference and flexible resource adjustments in the meantime. We will clarify this design in the revision, and will support more types of parallelism (e.g. pipeline parallelism) in the future.

---

> > ### Comment · Reviewer_j2me · 2025-08-05
> >
> > Thanks for this detailed explanation. I have no further concerns. Best,

---

### Official Review · Reviewer_WMmU · 2025-07-03

**Clarity:** 3
**Significance:** 3
**Originality:** 3
**Rating:** 4
**Confidence:** 3

**Summary:**

This paper introduces AGSERVE, a novel system designed to overcome the cost-quality tradeoff in serving Large Language Model (LLM) agents. Unlike traditional LLM serving systems, AGSERVE is optimized for the unique characteristics of agentic workflows, such as predictable request patterns, increasing quality requirements over multi-round sessions, and unique prompt formatting including middle truncation. AGSERVE features a session-aware server for efficient KV cache management, a quality-aware client that employs real-time quality assessment for dynamic model cascading, and a dynamic resource scheduler to maximize GPU utilization. Experimental results demonstrate that AGSERVE achieves comparable response quality to GPT-4o at a significantly lower cost (16.5% in some cases) , delivers a 1.8x improvement in quality relative to the cost-quality tradeoff curve , and increases KV cache hit rates by up to 2.86x compared to existing methods.

**Questions:**

- What are the specific heuristics or machine learning models used to predict the Estimated Time of Arrival (ETA) for sessions, which is crucial for the ECE eviction policy, and how robust are these predictions to unexpected agent behaviors or workload shifts?

- Beyond the qualitative "satisfactory rate drops as round increases" observation, can a more quantitative characterization of the increasing task difficulty for LLMs as context grows be provided, perhaps in terms of specific metrics or a more detailed analysis of failure modes at later stages of agent sessions?

- Given that the paper "disregard[s] the sessions that trigger rule violations"  when evaluating the performance of the RS Dynamic Allocation, how frequently do such violations occur in practice, what are their root causes, and how does AGSERVE handle them to maintain overall system stability and performance?

**Ethical Concerns:**

["NO or VERY MINOR ethics concerns only"]

**Final Justification:**

I've confirmed the author's rebuttal and will keep my original score.

**Limitations:**

Yes.

**Quality:**

3

**Strengths And Weaknesses:**

## Strengths

- The paper identifies and tackles the unique challenges of serving LLM agents, distinguishing it from traditional LLM serving systems that are not optimized for agent workflows. This focus on session-awareness, predictable request patterns, and growing context is a key strength.

- AGSERVE offers a holistic solution by integrating a session-aware server with innovative KV cache management (ETA-based eviction and in-place positional embedding calibration), a quality-aware client for session-aware model cascading, and a dynamic resource scheduler. These components work together to transcend the cost-quality tradeoff.

- The paper provides strong empirical evidence on real testbeds, showing that AGSERVE achieves comparable quality to GPT-4o at a substantially lower cost (16.5%), delivers a 1.8x quality improvement relative to the tradeoff curve, and significantly boosts KV cache hit rates (2.84x) while reducing latency.

## Weaknesses

- The evaluation primarily focuses on four specific agent implementations (AlfWorld, card game, knowledge graph, and Mind2Web) . While these cover different interaction patterns, the generalizability of AGSERVE's benefits across a wider range of LLM agent applications is not fully explored.

- The in-house deployment of AGSERVE primarily uses Llama-3 (8B and 70B versions) as the adjustable models . While GPT-4o is used as an external API , the system's performance and optimizations might be less effective or require significant re-tuning for other LLM architectures or model families that are not from the same family and do not share the same tokens and dictionary .

- The cost estimation for open-source models is based on retail cloud server rentals, and the paper acknowledges that "the cost of open-source models could be significantly lower (30% to 60% lower) for self-operated data centers or long-term users" . This simplification might not fully capture the true cost-efficiency advantages or disadvantages in all deployment scenarios, potentially skewing the cost-quality comparison.

- The R-Judge training relies on a customized Chatbot-Arena dataset where human experts labeled responses. The paper notes the difficulty of annotating prompt difficulty for Q-Judge , and it's unclear if the "human-selected preferences" for R-Judge fully capture the nuances of agent task success or are prone to subjective biases.

- While the paper mentions related works, some aspects of AGSERVE, like KV cache management and model cascading, could benefit from a more in-depth comparison with the very latest developments in those specific areas, beyond just general LLM serving systems. For instance, the discussion on cache reuse mentions CacheBlend , but a detailed comparative analysis of their respective strengths for agent serving is limited.

---

> ### Author Rebuttal · Authors · 2025-07-29
>
> Thanks for your detailed review, which helps us improve our quality of manuscript.
>
> [Weakness 1] The evaluation primarily focuses on four specific agent implementations . While these cover different interaction patterns, the generalizability of AGSERVE's benefits across a wider range of LLM agent applications is not fully explored.
> > Thanks for your advice. The four agents cover multiple interaction patterns, showing that AgServe improves agent serving efficiency with our proposed techniques based on agent serving behaviors. We’ll clarify AgServe’s generalization to agent serving scenarios in our revision, and expand agent implementation in the future.
>
> [Weakness 2] The system's performance and optimizations might be less effective or require significant re-tuning for other LLM architectures or model families that are not from the same family and do not share the same tokens and dictionary .
> > Thanks for your question. Generally speaking, LLMs with different dictionaries in the cascade will not affect AgServe’s performance.
> > 1. For generalizabilty, AgServe does not reuse tokenization result when performing service upgrade. Therefore, we expect little performacne divergence between heterogenous models and the current setup.
> > 2. Our profile on Llama-8B shows that tokenization takes only 1~2 milliseconds, while generation takes 2 seconds per round. Therefore, we believe the gain from dictionary reuse is limited.
> >
> > We‘ll clarify it in Appendix C.
>
> [Weakness 3] The simplification of VM rentals might not fully capture the true cost-efficiency advantages or disadvantages in all deployment scenarios, potentially skewing the cost-quality comparison.
> > Thanks for your question. AgServe keeps most lifetime of sessions on locally-deployed models, and only upgrades to GPT-4o when necessary. Therefore, lower rental price is beneficial for AgServe. The table below shows the performance of AgServe for Figure 8(a) if rental is 30% or 60% lower. We’ll add these statistics to Appendix D.
>
> ||AgServe|Llama-8B|Llama-70B|GPT-4o|RouteLLM|Cascade|
> |-|-|-|-|-|-|-|
> |Original Cost|0.0177|0.0023|0.0376|0.1073|0.0337|0.1395|
> |Cost(-30\% Rent)|0.0137|0.0016|0.0263|0.1073|0.0324|0.1353|
> |Cost(-60\% Rent)|0.0097|0.0009|0.0151|0.1073|0.0312|0.1311|
> |Quality|90|21|72|94|25|69|
>
> [Weakness 4] It's unclear if the "human-selected preferences" for R-Judge fully capture the nuances of agent task success or are prone to subjective biases.
> > Thanks for your question. To minimize the bias, the annotators were shown the observation and LLM’s response round by round, with no previous knowledge of the session’s final result or the model size. We also utilize the voting mechanism and each label is agreed by three annotators at least. We’ll clarify it in our revision.
>
> [Weakness 5] Some aspects of AGSERVE, like KV cache management and model cascading, could benefit from a more in-depth comparison with the very latest developments in those specific areas, beyond just general LLM serving systems. For instance, the discussion on cache reuse mentions CacheBlend , but a detailed comparative analysis of their respective strengths for agent serving is limited.
> > Thanks for your advice. As we discussed in Appendix F, CacheBlend is orthogonal to AgServe. CacheBlend selectively updates the KV cache for pre-computed RAG contents, and is applicable to AgServe. We will discuss CacheBlend in the main text in the revised version, if the space permits.
>
> [Question 1] What are the specific heuristics or machine learning models used to predict the Estimated Time of Arrival (ETA) for sessions, which is crucial for the ECE eviction policy, and how robust are these predictions to unexpected agent behaviors or workload shifts?
> > Thanks for your insight. AgServe predicts ETA in the following manner. First, AgServe maintains the metadata of each session. When a session starts, AgServe initialize the metadata by the agent type. Throughout the session, AgServe updates the metadata with the last several rounds’ round duration, and last arrival time data. AgServe will use the average round duration to calculate the ETA. The ETA grows if the next request has not arrived at the original ETA time, and finally reaches infinite in case of unexpected session interruption. The ETA prediction will not be drastically affected by the unexpected behaviors, since it keeps track of multiple rounds of history data. It is robust to workload shift, since each session is treated independently.
>
> [Question 2] Beyond the satisfactory rate observation, can a more quantitative characterization of the increasing task difficulty for LLMs as context grows be provided, perhaps in terms of specific metrics or a more detailed analysis of failure modes at later stages of agent sessions?
> > Thanks for your question. Task difficulty is a subjective metric, which is hard to quantify. We discussed different types of quality issues in Section 5.2 (L139), which can be considered as different modes of failures. Taking Alfworld as an example, here is a detailed analysis of agent outcomes for 20 AW cases. There is an increasing trend for invalid action or format in the later stages and smaller models. We’ll add more discussions on failure modes in our revision of Section 2.2.
>
>
> ||Success|Reach Round Limit|Invalid Actions|Rule Violation|
> |-|-|-|-|-|
> |Llama-8B|35%|40%|15%|10%|
> |Llama-70B|65%|30%|5%|0%|
>
> [Question 3] Given that the paper "disregard[s] the sessions that trigger rule violations" in Sec 7.4.3, how frequently do such violations occur in practice, what are their root causes, and how does AGSERVE handle them to maintain overall system stability and performance?
> > Thanks for your question. These violations mainly refer to the violation of preset agent-specific rules or action format. This will make agent exit prior to task completion or reaching round limit. Such violation takes place in 6% of sessions. This probability is similar for FS1, FS2 and AgServe. This is mainly due to the LLM’s limited capability in long-context scenarios. In its normal operation, AgServe will choose to upgrade the model to GPT-4o.

---

### Official Review · Reviewer_Z6NR · 2025-07-07

**Clarity:** 3
**Significance:** 3
**Originality:** 3
**Rating:** 4
**Confidence:** 4

**Summary:**

The paper presents a session-aware system for LLM agent serving, addressing inefficiencies in KV cache management, model selection, and resource allocation. Key innovations include ETA-based KV cache eviction, in-place positional embedding calibration for middle-truncation reuse, quality-aware cascading via QMM, and dynamic GPU scheduling. Evaluated on AgentBench with agents (e.g., ALFWorld, Mind2Web) and models (Llama-3 8B/70B, GPT-4o), it achieves 83.5% cost reduction at GPT-4o quality, 1.8x quality at same cost, 2.84x cache hit rate, and up to 50% latency cut vs. baselines like vLLM and Llumnix++.

**Questions:**

- The authors mentioned the long-input scenarios in sec 7.2. Could the authors elaborate a bit more on how to handle the straggler settings with the four agents?
- Could the authors provide more cascade performance with more levels/non-Llama models, and potentially extended to large-scale serving involving more than two nodes?

**Ethical Concerns:**

["NO or VERY MINOR ethics concerns only"]

**Final Justification:**

I appreciate the clarification and new experiments addressing other reviewers' comments. I have decided to keep my original ratings.

**Limitations:**

Yes

**Quality:**

3

**Strengths And Weaknesses:**

Strength:
- The paper is clearly presented and well written.
- The paper proposes the session-aware kv cache strategy, the proposed eta eviction and calibration enhance reuse for multi-round agents, outperforming LRU with 2.84x hit rates. This could offer tailored advantages against existing prefix-focused systems, which optimize general inference but overlook agent-specific middle-truncation.
- The paper proposes quality-aware cascading, the session-level upgrades via QMM (95.2% recall) minimize costs by preferring underkill, surpassing RouteLLM, and could achieve similar quality to GPT-4o with a 16.5% cost in AW.
- The proposed system integrates cache, cascading, and scheduling for 65.6% multi-agent savings and provides throughout empirical evaluations on AgentBench and synthetic request traces from Poisson and Gamma distributions. Theoretical proofs has also been provided for in-place kv-cache calibration.

Weakness:
- The assumption of ece is uniform/burst, which may be vulnerable to the variability as noted in [1].
- In sec 7.4.2, the authors discuss the low overhead of QMM, but this may accumulate with long-input or long-context checks (different scenarios of prefilling/decoding bound), providing more details about the context-length, quality & efficiency tradeoff could make the claim stronger;
- It is understandable that the study already provides throughout analyses on A6000 and recent AgentBench, but the scalability might be untested vs. heterogeneous LLMs and the fixed three-tier cascade lacks diversity;

[1] Wang, Jiahao, et al. "KVCache Cache in the Wild: Characterizing and Optimizing KVCache Cache at a Large Cloud Provider." arXiv preprint arXiv:2506.02634 (2025).

---

> ### Author Rebuttal · Authors · 2025-07-29
>
> Thank you for your postive appraisal of our work!
>
> [Weakness 1] The assumption of ece is uniform/burst, which may be vulnerable to the variability as noted in [1].
> > Thanks for bringing [1] to our attention. ECE relies on the assumption that the estimated time of arrival (ETA) for the next request in a session is independently predictable. This assumption holds well in practice for agent workloads.
> >
> > Interestingly,  as a contemporaneous work, [1] investigates multi-turn request traces (particularly agent workloads as cited by their [55, 57]) and arrives at a similar conclusion. As shown in their Section 3.3,  _the probability of issuing the next turn request is predictable given a workload category and time interval_, which supports our design. We will explicitly discuss and cite [1] in Section 2.2 of our revision to acknowledge this connection.
>
> [Weakness 2] The low overhead of QMM may not work under accumulated long contexts.
> > Thanks for your question. The two judges’ context window is 512, limited by their backbone (i.e. BERT, DistilBERT). Q-Judge's input of $t$ and $p$ is usually long enough to saturate the window. As for R-Judge, we feed as much information as possible in $e$ to fill the window up, since longer context facilitates R-Judge to capture the LLM’s repetitive behavior if applicable. As a result, AgServe almost always utilize full context window of both judges in our evaluation. Despite that, we witness no significant overhead, since QMM overlaps with observation generation. We will clarify it in Section 7.4.2 of our revised manuscript.
>
> [Weakness 3, Question 2] Could the authors provide more cascade performance with more levels/non-Llama models, and potentially extended to large-scale serving involving more than two nodes?
> > Thanks for your question. We will address your concern in three parts.
> > 1. AgServe selects the three-tier cascade due to the capability gap of LLMs and the limited testbed size, justified in Appendix C (L639). We will add reference to it in the main body.
> > 2. Llama structure is adopted by many other LLMs, such as the Qwen family. Therefore, these models benefit the same from SAS's technology just like Llama models. Moreover, AgServe does not take advantage from tokenization reuse, as tokenization only takes negligible time of 1~2 milliseconds compared to the 2 second decoding duration for Llama-8B. Therefore, heterogenous models will not affect AgServe's performance.
> > 3. Since each SAS processes its sessions independently in AgServe, we expect little performance divergence for larger testbeds. We are working on finding a larger testbed, and we expect to evaluate on a larger scale in the revised version.
>
> [Question 1] The authors mentioned the long-input scenarios in sec 7.2. Could the authors elaborate a bit more on how to handle the straggler settings with the four agents?
> > Thanks for your question. We would like to explain why we claim that RouteLLM can not tackle the long-input scenarios. RouteLLM is tailored for short-input applications (e.g. MMLU[1] and GSM8K[2]), where average input is less than 100 tokens. However, in general, the input for agent serving exceeds thousand tokens due to the long few-shot examples and tool instructions. To tackle this problem, AgServe's QMM selectively picks the information that are necessary for quality assessment. This enables QMM to perform well in the long-input agent serving scenarios. We will clarify it in the revision.
> >
> > [1] D.Hendrycks, _et al_. Measuring massive multitask language understanding. ICLR 2020.
> >
> > [2] K.Cobbe, _et al_. Training verifiers to solve math word problems. arXiv:2110.14168, 2021.

---

### Note · Authors · 2025-08-12

Dear AC and reviewers,

We thank the AC and reviewers for their valuable feedback and constructive comments. In this work, we present AgServe, the first serving system specifically designed for agent workflows. As reviewer WMmU aptly summarizes, AgServe provides a holistic solution by integrating a session-aware server with innovative KV cache management, a quality-aware client for session-based model cascading, and a dynamic resource scheduler. These components work synergistically to go beyond the traditional cost–quality trade-off.

During the rebuttal period, we provided the following explanation and evidence to support our manuscript:
﻿
1. AgServe’s assumptions about agent serving patterns are consistent with contemporaneous work.
2. The ECE algorithm and QMM module remain highly efficient even under extreme conditions.
3. AgServe’s architecture is compatible with multiple models and does not require cascades to belong to a single model family.
4. AgServe consistently outperforms the quality–cost curve under various rental pricing plans.
5. AgServe’s resource allocation is scalable and can leverage tensor parallelism.
6. We have clarified other concerns raised by the reviewers, and will revise the manuscript accordingly to ensure clearer presentation.

We sincerely thank the reviewers for their time and for accepting our responses. We will ensure that all feedback is fully incorporated into the revision, and will continue to improve AgServe to support an even broader range of implementations.

---

### Decision · Program_Chairs · 2025-09-17

**Decision:**

Accept (poster)

**Comment:**

The paper presents an agentic and session-aware KV-cache management system. The system leads to better GPU utilization without compromising the accuracy and quality of responses. All the reviewers give 'borderline accept' ratings to the paper. Some of them increased their initial ratings after reading the authors' rebuttal and additional experimental results. The reviewers expressed some concerns about the writing and clarity of the paper, which I encourage the authors to address in the revised version of the paper.